# dFLASH; dual FLuorescent transcription factor activity sensor for histone integrated live-cell reporting and high-content screening

Timothy P. Allen [1], Alison E. Roennfeldt[1,2], Moganalaxmi Reckdharajkumar [1], Adrienne E. Sullivan [3,4], Miaomiao Liu [5], Ronald J. Quinn [5,7], Darryl L. Russell[2,7], Daniel J. Peet [1,7], Murray L. Whitelaw[1,6,7] & David C. Bersten [1,2] ✉

Live-cell transcription factor (TF) activity reporting is crucial for synthetic biology, drug discovery and functional genomics. Here we present dFLASH (dual FLuorescent transcription factor Activity Sensor for Histone-integrated live-cell reporting), a modular, genome-integrated TF sensor. dFLASH homogeneously and specifically detects endogenous Hypoxia Inducible Factor (HIF) and Progesterone Receptor (PGR) activities, as well as coactivator recruitment to synthetic TFs. The dFLASH system produces dual-color nuclear fluorescence, enabling normalized, dynamic, live-cell TF activity sensing with strong signal-to-noise ratios and robust screening performance ($Z' = 0.61-0.74$). We validate dFLASH for functional genomics and drug screening, demonstrating HIF regulation via CRISPRoff and application to whole-genome CRISPR KO screening. Additionally, we apply dFLASH for drug discovery, identifying HIF pathway modulators from a 1600-compound natural product library using high-content imaging. Together, this versatile platform provides a powerful tool for studying TF activity across diverse applications.

Cells integrate biochemical signals in a variety of ways to mediate effector function and alter gene expression. Transcription factors (TF) sit at the heart of cell signalling and gene regulatory networks, linking environment to genetic output[1,2]. TF importance is well illustrated by the consequences of their dysregulation within disease, particularly cancer where TFs drive pathogenic genetic programmes[3–5]. As a result, there is widespread utility in methods to manipulate and track TF activity in basic biology and medical research, predominantly using TF responsive reporters. Recent examples include enhancer activity screening[6] by massively parallel reporter assays, discovery and characterisation of transcription effector domains[7,8] and CRISPR-based functional genomic screens that use reporter gene readouts to understand transcriptional regulatory networks[2,9]. Beyond their use in discovery biology, TF reporters are increasingly utilised as sensors and actuators in engineered synthetic biology applications such as diagnostics and cellular therapeutics. For example, synthetic circuits that

[1]School of Biological Sciences, University of Adelaide, Adelaide, SA, Australia. [2]Robinson Research Institute, University of Adelaide, Adelaide, SA, Australia. [3]Adelaide Centre for Epigenetics, School of Biomedicine, Faculty of Health and Medical Sciences, The University of Adelaide, Adelaide, SA, Australia. [4]South Australian immunoGENomics Cancer Institute (SAiGENCI), Faculty of Health and Medical Sciences, The University of Adelaide, Adelaide, SA, Australia. [5]Griffith Institute for Drug Discovery, Griffith University, Brisbane, QLD, Australia. [6]ASEAN Microbiome Nutrition Centre, National Neuroscience Institute, Singapore 308433, Singapore. [7]These authors contributed equally: Ronald J. Quinn, Darryl L. Russell, Daniel J. Peet, Murray L. Whitelaw. ✉e-mail: david.bersten@adelaide.edu.au

utilise either endogenous or synthetic TF responses have been exploited to engineer cellular biotherapeutics[10]. In particular, the synthetic Notch receptor (SynNotch) in which programable extracellular binding elicits synthetic intracellular TF signalling has been used to enhance tumour-specific activation of CAR-T cells, overcome cancer immune suppression, or provide precise tumour target specificity[11–14].

Fluorescent reporter systems are now commonplace in many studies, linking cell signalling to TF function and are particularly useful to study single cell features of gene expression, such as stochastics and heterogeneity[15], or situations where temporal recordings are required. In addition, pooled CRISPR/Cas9 functional genomic screens rely on the ability to select distinct cell pools from a homogenous reporting parent population. Screens to select functional gene regulatory elements or interrogate chromatin context in gene activation also require robust reporting in polyclonal pools[16]. Many of the current genetically encoded reporter approaches, by nature of their design, are constrained to particular reporting methods or applications[9,17]. For example, high content arrayed platforms are often incompatible with flow cytometry readouts and vice versa. As such, there is a need to generate modular, broadly applicable platforms for robust homogenous reporting of transcription factor and molecular signalling pathways[2].

Here we address this by generating a versatile, high-performance sensor of signal regulated TFs. We develop a reporter platform, termed the dual FLuorescent TF Activity Sensor for Histone integrated live-cell reporting (dFLASH). dFLASH is a TF responsive reporter coupled to an internal control with lentiviral mediated genomic integration. The modular composition of the dFLASH cassette is critical to the robust enhancer-driven reporting achieved, which we demonstrate using the well-defined hypoxic and steroid receptor signalling pathways. dFLASH acts as a dynamic sensor of targeted endogenous pathways as well as synthetic TF chimeras in polyclonal pools by temporal high-content imaging and flow cytometry. Routine isolation of homogenously responding monoclonal reporter lines enabled robust high content image-based screening ($Z' = 0.61$-$0.74$) for signal regulation of endogenous and synthetic TFs, as well as demonstrating utility for functional genomic investigations using CRISPRoff and whole genome pooled CRISPR KO screens. Drug screening using array-based temporal high content imaging with a hypoxia inducible factor responsive dFLASH reporter successfully identified previously unknown regulators of the hypoxic response pathway, illustrating the suitability of dFLASH for arrayed drug screening applications. Together, we show that the dFLASH platform allows for intricate interrogation of signalling pathways and illustrate its value for functional gene discovery, evaluation of regulatory elements and investigation into chemical targeting of TF regulation.

## Results

### Design of dFLASH, a dual fluorescent, live cell sensor of TF activity

To fulfil the need for a versatile fluorescent sensor cassette that can be integrated into chromatin and enable robust live-cell sensing, a lentiviral construct with enhancer regulated expression of Tomato, followed by independent, constitutive expression of d2EGFP as both a selectable marker and an internal control was constructed (Fig. 1a, b). Three nuclear localisation signals (3xNLS) integrated in each fluorescent protein ensured nuclear enrichment to enable single cell identification by nuclear segmentation in high content image analysis, which can be coupled with imaged-based quantification for normalised reporter output. Additionally, FACS can be used for single-cell isolation in a signal dependent or independent manner. The enhancer cassette upstream of the low background optimised minimal promoter[18] driving Tomato expression is flanked by restriction sites, enabling alternative enhancer cloning (Fig. 1a). This design is adaptable

for any nominated TF, applicable to high content imaging (HCI) and can be used for the selection of single responding cells from polyclonal pools via image segmentation or flow cytometry (Fig. 1c).

dFLASH response to endogenous signal-regulated TF pathways was first assessed by inserting a Hypoxia Inducible Factor (HIF) responsive enhancer element. HIF-1 is the master regulator of cellular adaption to low oxygen tension and has important roles in several diseases[19–21]. To mediate its transcriptional programme, the HIF-1α subunit heterodimerises with Aryl Hydrocarbon Nuclear Translocator (ARNT), forming an active HIF-1 complex. At normoxia[4], HIF-1α is post-translationally downregulated through the action of prolyl hydroxylase (PHD) enzymes and the Von Hippel Lindau (VHL) ubiquitin ligase complex[22]. Additionally, the C-terminal transactivation domain of HIF-1α undergoes asparaginyl hydroxylation mediated by Factor Inhibiting HIF (FIH), which blocks binding of transcription coactivators CBP/p300[23]. These hydroxylation events are repressed during low oxygen conditions, enabling rapid accumulation of active HIF-1α. HIF-1α stabilisation at normoxia[4] can be artificially triggered by treating cells with the hypoxia mimetic dimethyloxalylglycine (DMOG), which inhibits the PHDs and FIH, thereby inducing HIF-1α stabilisation, activity and hypoxic gene expression[24]. The well characterised regulation and disease relevance of HIF-1 made it an ideal TF target for prototype sensor development.

### Optimisation of dFLASH sensors

Initially, we tested dFLASH constructs with repeats of hypoxia response element (HRE) containing enhancers (RCGTG) from well described endogenous target genes *Pgk1*, *Eno1* & *Ldha* (HRE-FLASH, no GFP internal control). In these constructs, the HRE enhancer controls expression of either nuclear mono (m) or tandem dimer (td)Tomato, however no DMOG induced Tomato expression was observed in stable HEK293T cell lines (mnucTomato or tdnucTomato, Supplementary Fig. 1a, b). Given the HIF response element has been validated previously[25], the response to HIF-1α was optimised by altering the reporter design, all of which utilised the smaller mnucTomato (vs tdnucTomato) to restrain transgene size. We hypothesised that transgene silencing, chromosomal site-specific effects or promoter enhancer coupling/interference may result in poor signal induced reporter activity observed in initial construct designs. As such, we optimised the downstream promoter, the reporter composition and incorporated a 3xNLS d2EGFP internal control from the constitutive promoter to monitor chromosomal effects and transgene silencing.

Dual FLASH (dFLASH) designs incorporated three variations of the downstream promoter (EF1α, PGK and PGK/CMV) driving *3xNLS EGFP (nucEGFP)* and 2A peptide linked hygromycin (detailed in Supplementary Fig. 1c) in combination with alternate reporter transgenes that expressed mnucTomato alone, or mnucTomato-Herpes Simplex Virus Thymidine Kinase (HSVtK)-2A-Neomycin resistance gene (Neo). Stable HEK293T and HepG2 HRE-dFLASH cell lines with these backbones were generated by lentiviral transduction and hygromycin selection. The reporter efficacy of dFLASH variant cell lines was subsequently monitored by high content imaging 48 h after DMOG induction (Supplementary Fig. 1d, e). The downstream composite PGK/CMV or PGK promoters, resulted in strong DMOG induced Tomato or Tomato/GFP expression which dramatically outperformed EF1α (Fig. 1b and Supplementary Fig. 1d). The composite PGK/CMV provided bright, constitutive nucEGFP expression in both HepG2 and HEK293T cells which was unchanged by DMOG, whereas nucEGFP controlled by the PGK promoter was modestly increased (~2.5 fold) by DMOG (Supplementary Fig. 1e). Substitution of the mnucTomato with the longer mnucTomato-HSVtK-Neo reporter had no effect on DMOG dependent reporter induction in EF1α containing HRE-dFLASH cells, which still failed to induce tomato expression (Supplementary Fig. 1f). CMV/PGK containing dFLASH sensors maintained DMOG induction when either the mnucTomato or the mnucTomato/HSVtK/Neo

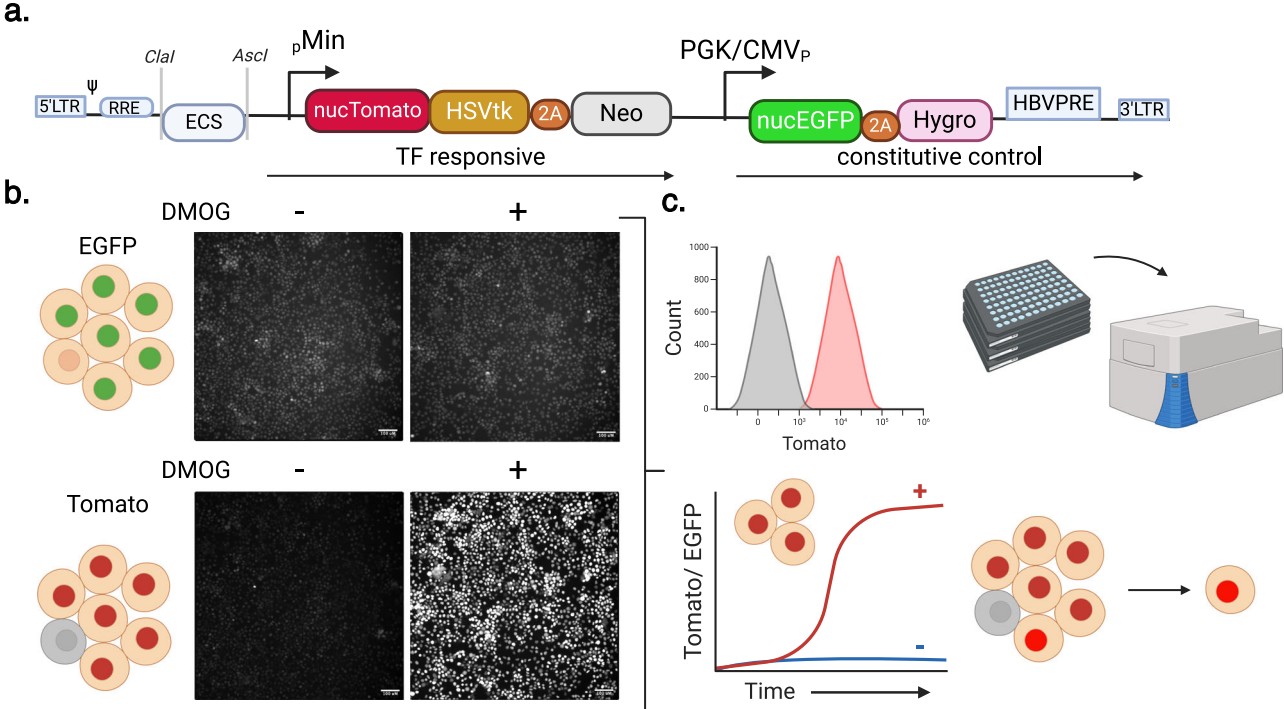

**Fig. 1 | Summary of dFLASH LV-REPORT construction, utility, and validation.**
**a** The dFLASH system utilises the lentiviral LV-REPORT construct, consisting of a cis-element multiple cloning site (ECS) for enhancer insertion (flanked by ClaI and AscI restriction enzyme sites), followed by a minimal (min) promoter that drives a transcription factor (TF) dependent cassette that encodes three separate expression markers; a nuclear Tomato fluorescent protein with a 3x C-terminal nuclear localisation signal (NLS), Herpes Simplex Virus Thymidine Kinase (HSVtK) for negative selection and, separated by a 2A self-cleaving peptide (2A), Neomycin resistance gene (Neo) for positive selection. This is followed by a downstream constitutive promoter (PGK/CMV) that drives an independent cassette encoding EGFP with a 3x N-terminal NLS and, separated by a 2A peptide, a Hygromycin (Hygro) resistance selection marker. This construct is flanked either side by long terminal repeats (LTR) for lentiviral mediated insertion. **b** This design allows for initial identification of the EGFP fluorescent protein in nuclei, independent of TF-dependent signal. Expression of the Tomato fluorescent protein is highly upregulated in a signal-dependent manner. Images shown are monoclonal HEK293T dFLASH-HIF cells. Populations were treated for 48 h ± DMOG to induce HIF-1α and were imaged by HCI. Data representative of $n = 3$ independent experiments. Scale bar = 100 µM. **c** This system can be adapted to a range of different applications. This includes (clockwise) flow cytometry, arrayed screening in a high throughput setting with high content imaging, isolation of highly responsive clones or single cells from a heterogenous population or temporal imaging of pooled or individual cells over time. Created in BioRender. Peet, D. (2025) https://BioRender.com/b38b268.

reporters were utilised (Supplementary Fig. 1g, h) although mnucTomato without HSVtK and Neo produced lower absolute mnucTomato fluorescence and a smaller percentage of cells responding to DMOG, albeit with lower background.

Taken together these findings indicate that certain backbone compositions prevented or enabled robust activation of the enhancer driven cassette, similar to the suppression of an upstream promoter by a downstream, contiguous promoter previously described[26,27] suggesting that the 3' EF1α promoter results in poorly functioning multicistronic synthetic reporter designs[28]. Consequently, the PGK/CMV backbone and the mnucTomato/HSVtK/Neo reporter from Supplementary Fig. 1 was chosen as the optimised reporter design (HRE-dFLASH).

To confirm that the HRE element was conferring HIF specificity, a no response element dFLASH construct in HEK293T cells treated with DMOG produced no change in either mnucTomato or nucEGFP compared to vehicle-treated populations (supplementary Fig. 2a). This result, together with the robust induction in response to DMOG (Fig. 2D, Supplementary Fig. 1f, h), confirms the HIF enhancer driven reporter responds robustly to induction of the HIF pathway (subsequently referred to as dFLASH-HIF).

To validate the high inducibility and that the nucEGFP independence of dFLASH was not restricted to the HIF pathway, we generated a Gal4 responsive dFLASH construct (Gal4RE-dFLASH), using Gal4 responsive enhancers[23,29]. HEK293T cells were transduced with Gal4RE-

dFLASH and a dox-inducible expression system to express a synthetic Gal4DBD-transactivation domain fusion protein. To evaluate Gal4RE-dFLASH, we expressed Gal4DBD fused with a compact VPR (miniVPR), a strong transcriptional activator[30] (supplementary Figs. 2b, 4a–c). We observed ~25% of the polyclonal population was highly responsive to doxycycline treatment (Supplementary Fig. 2b), with a ~14-fold change in Tomato expression relative to nucEGFP by HCI (Supplementary Fig. 4c), demonstrating our optimised dFLASH backbone underpins a versatile reporting platform.

### dFLASH senses functionally distinct TF activation pathways
Following the success in utilising dFLASH to respond to synthetic transcription factor and HIF signalling, we explored the broader applicability of this system to sense other TF activation pathways. We chose the Progesterone Receptor (PGR), a member of the 3-Ketosteriod receptor family that includes the Androgen, Glucocorticoid and Mineralocorticoid receptors, as a functionally distinct TF pathway with dose-dependent responsiveness to progestin steroids, to assess the adaptability of dFLASH performance. Keto-steroid receptors act through a well-described mechanism which requires direct ligand binding to initiate homodimerization via their Zinc finger DNA binding domains, followed by binding to palindromic DNA consensus sequences. PGR is the primary target of progesterone (P4, or a structural mimic R5020) and has highly context dependent roles in reproduction depending on tissue type[31–33]. We inserted PGR-target gene

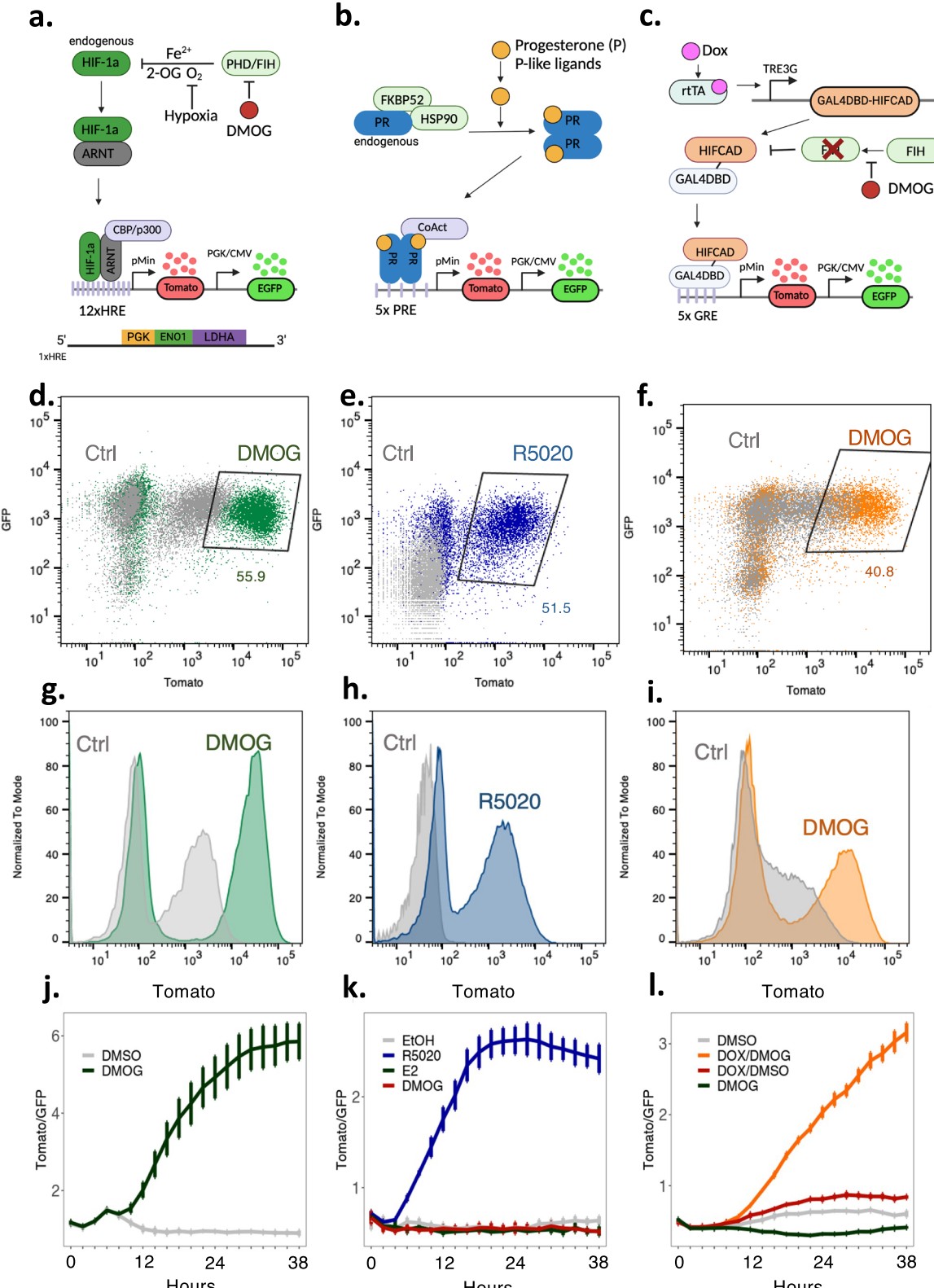

enhancer sequences containing the canonical NR3C motif (ACANNNTGT[32]) into dFLASH, conferring specificity to the ketosteroid receptor family to generate PRE-dFLASH (Fig. 2b, see Methods).

A chimeric TF system was also established using Gal4DBD fusion proteins to create a synthetic reporter to sense the enzymatic activity of the oxygen sensor Factor Inhibiting HIF (FIH). This sensor system, termed SynFIH for its ability to synthetically sense FIH activity, contained a Gal4DBD-HIFCAD fusion protein expressed in a doxycycline-dependent manner, in cells harbouring stably integrated Gal4RE-dFLASH. FIH blocks HIF transactivation through hydroxylation of a conserved asparagine in the HIF-1α C-terminal transactivation domain (HIFCAD), preventing recruitment of the CBP/p300 co-activator complex[23]. As FIH is a member of the 2-oxoglutarate dioxygenase family, like the PHDs which regulate HIF post-translationally, it is also inhibited by DMOG (Fig. 2C), allowing induction of SynFIH-dFLASH upon joint Dox and DMOG treatment (Supplementary Fig. 4d, e).

**Fig. 2 | dFLASH provides a sensitive readout of three distinct TF pathways.**
**a**–**c** Three distinct enhancer elements enabling targeting of three different signalling pathways. **a** Hypoxic response elements (HRE) provide a read out for HIF-1α activation (2OG; 2-oxogluterate, PHD; prolyl hydroxyl domain protein, FIH; factor inhibiting HIF); **b** Progesterone response elements (PRE), derived from progesterone receptor target genes, facilitate reporting of progestin signalling (FKBP52, HSP90 heat shock proteins, PR progesterone receptor, CoAct transcriptional coactivators) (**c**) Gal4 response elements (GRE) enable targeting of synthetic transcription factors to dFLASH such as a GAL4DBD-HIFCAD fusion protein that provides a FIH-dependent reporter response (rtTA; reverse tetracycline-controlled transactivator, gal4DBD; gal4 DNA binding domain, HIFCAD; HIF1-α C-terminal transactivation domain). **a**–**c** Created in BioRender. Peet, D. (2025) https://BioRender.com/s67t615 **d**–**i** Flow cytometry histograms and dot plots showing Tomato expression following 48 h treatments of the indicated dFLASH polyclonal reporter cells (**d, g**) HEK293T HRE-dFLASH; 1 mM DMOG or 0.1% DMSO (ctrl), (**e, h**) T47D PRE-dFLASH; 100 nM R5020 or Ethanol (ctrl), (**f, i**) HEK293T dFLASH-synFIH; 1 μg/mL Doxycycline (Dox) and 1 mM DMOG or Dox and 0.1% DMSO (ctrl). Percentages of Tomato-positive ligand-treated population are displayed. **j**–**l** Reporter populations as in (**d**–**i**) were temporally imaged for 38 h using HCI directly after treatment with (**j**) 0.5 mM DMOG or 0.1% DMSO, (n = 4 biological replicates per group) (**k**) 100 nM R5020, 35 nM E2, 0.5 mM DMOG or 0.1% Ethanol (EtOH) (n = 4 biological replicates per group) (**l**) 0.1% DMSO, 1 mM DMOG, 100 ng/mL Dox and 0.1% DMSO, or 100 ng/mL Dox and 1 mM DMOG (n = 4 biological replicates per group). Data presented as mean ± sem in (**j**–**l**). Source data are provided as a Source Data file.

dFLASH-based sensors for PGR and Gal4DBD-HIFCAD were generated in the optimised backbone used for dFLASH-HIF (Fig. 2a–c). For the PGR sensor, we transduced T47D cells with PRE-dFLASH, as these have high endogenous PGR expression, while for the FIH-dependent system we generated HEK293T cells with Gal4RE-dFLASH and the Gal4DBD-HIFCAD constructs by lentiviral transduction (dFLASH-synFIH).

Stable polyclonal cell populations were treated with their requisite chemical regulators and reporter responses analysed by either flow cytometry or temporal imaging using HCI at 2 h intervals for 38 h (Fig. 2). Flow cytometry revealed all three systems contain a population that strongly induced nucTomato and maintained nucEGFP (Supplementary Fig. 2). In HEK293T cells, ~41% of the dFLASH-synFIH and ~56% of the dFLASH-HIF populations induced Tomato fluorescence substantially relative to untreated controls (Fig. 2d, f). The ~41% reporter response to inhibited FIH activity by DMOG (Supplementary Fig. 2e, Fig. 2f) is comparable with what was observed for Gal4RE-dFLASH response to Gal4DBD-miniVPR expression after equivalent selection (Supplementary Fig. 2b). The PGR reporter in T47D cells showed ~50% of the population substantively induced Tomato (Fig. 2e, Supplementary Fig. 2d). The presence of considerable responsive populations for FIH (~41%), PGR (~52%), and HIF sensors (~56%), reflected in the histograms of the EGFP positive cells (Fig. 2g-i) indicated that isolation of a highly responsive clone or subpopulations can be readily achievable for a range of transcription response types. Importantly, in FIH (KO) dFLASH-synFIH cells, the induction of dFLASH-synFIH by Dox/DMOG co-treatment was ablated and instead high basal Tomato expression was observed (Supplementary Fig. 4e), indicating that the dFLASH-synFIH system specifically senses FIH enzymatic activity.

All dFLASH systems showed consistent signal-dependent increases in reporter activity out to 38 h by temporal HCI, demonstrating that polyclonal populations of dFLASH can be used to track TF activity (Fig. 2j–l). PRE-dFLASH was more rapidly responsive to R5020 ligand induction (~6 h, Fig. 2k) than dFLASH-HIF and dFLASH-synFIH to DMOG or Dox/DMOG treatment, respectively (~10 h, Fig. 2j, l). Treatment of PRE-dFLASH with oestrogen (E2), which activates the closely related Estrogen Receptor, or the hypoxia pathway mimetic DMOG, failed to produce a response on PRE-dFLASH (Fig. 2k). This indicates that the PRE enhancer element is selective for the ketosteroid receptor family (also see below), and that enhancer composition facilitates pathway specificity. We also observed a signal-dependent change in EGFP expression by flow cytometry in the T47D PRE-dFLASH reporter cells (Supplementary Fig. 2g) but did not observe a significant change in EGFP expression for HEK293T or HEPG2 dFLASH-HIF (Supplementary Fig. 1e, Supplementary Fig. 2f) or in HEK293T dFLASH-synFIH cells (Supplementary Fig. 2h), with only a small change in dFLASH-synVPR cells (Supplementary Fig. 2b). While this change in T47D cells was not detected in the other cellular contexts (see below), it highlights that care needs to be taken in confirming the utility of the constitutive nucEGFP as an internal control in certain scenarios. The dFLASH system was also amenable for use in primary non-immortalised cells, demonstrated using mouse granulosa cells (Supplementary Fig. 3a–e).

We introduced dFLASH-HRE lentivirus into primary mouse granulosa cells and stimulated cells with the PHD inhibitor FG-4592. Both high content imaging (Supplementary Fig. 3a–c) and flow cytometry (Supplementary Fig. 3d, e) detected significant induction of Tomato fluorescence, however similar to the T47D PRE-dFLASH line, EGFP expression changed with FG-4592 treatment. Thus, confirmation of the use of EGFP as an internal control needs to be initially assessed in a primary cell context.

## Monoclonal dFLASH cell lines confer robust screening potential in live cells

The observed heterogenous expression of dFLASH within polyclonal cell pools is useful in many assay contexts but reduces efficiency in arrayed high content screening experiments and is incompatible with pooled isolation of loss of function genetic regulators. Therefore, monoclonal (mc) HEK293T and HepG2 dFLASH-HIF, T47D and BT474 PRE-dFLASH and HEK293T dFLASH-synFIH cell lines were derived to increase reliability of induction, as well as consistency and homogeneity of reporting (Fig. 3, Supplementary Fig. 5). The isolated mcdFLASH-synFIH and mcdFLASH-HIF lines also maintained constitutive, signal insensitive nucEGFP expression (Supplementary Fig. 5a, b, h). While T47D PRE-mcdFLASH showed a small increase in nucEGFP in response to R5020 (Supplementary Fig. 5d), this did not preclude its use in normalisation of high content imaging experiments (see below). However, no change in EGFP was observed in the BT474 PRE-mcdFLASH line (Supplementary Fig. 5e), suggesting that in the T47D PRE-mcdFLASH line, cell-type specific effects may be leading to R5020 dependent changes in EGFP. Flow cytometry of monoclonal dFLASH cell lines with their cognate ligand inducers (DMOG (Fig. 3b), R5020 (Fig. 3h), or Dox/DMOG (Fig. 3n)) revealed robust, homogeneous induction of mnucTomato in all cell lines. Specifically, mcdFLASH-HRE displayed Tomato induction in ~95% (HEK293T, Supplementary Fig. 5a) and 90% (HepG2, Supplementary Fig. 5b) of cells, mcdFLASH-PRE displayed tomato induction in ~95% (T47D, Supplementary Fig. 5d) and 96% (BT474, Supplementary Fig. 5e) of cells, and HEK293T mcdFLASH-SynFIH induced Tomato in ~81% of cells (Supplementary Fig. 5h).

Using temporal high content imaging we also found that clonally derived lines displayed similar signal induced kinetics as the polyclonal reporters, although displayed higher signal to noise and increased consistency (Fig. 3, Supplementary Fig. 5). We further confirmed that the HRE-dFLASH system is sensitive to physiological stimuli in HEK293T mcdFLASH-HIF cells, demonstrating a homogenous response after 24 h 1% $O_2$ hypoxic stimulation (Fig. 3f,g). In addition, using physiologically relevant concentrations of steroids or steroid analogues (10 nM-35 nM), the PRE-mcdFLASH lines selectively responds to R5020 (10 nM) but not E2 (35 nM), DHT (10 nM), Dexamethasone (Dex, 10 nM) or Retinoic acid (RA, 10 nM) (Fig. 3i, m, Supplementary Fig. 5k). Dose response curves of R5020 mediated Tomato induction indicate that the PRE-mcdFLASH line responds to R5020 with an

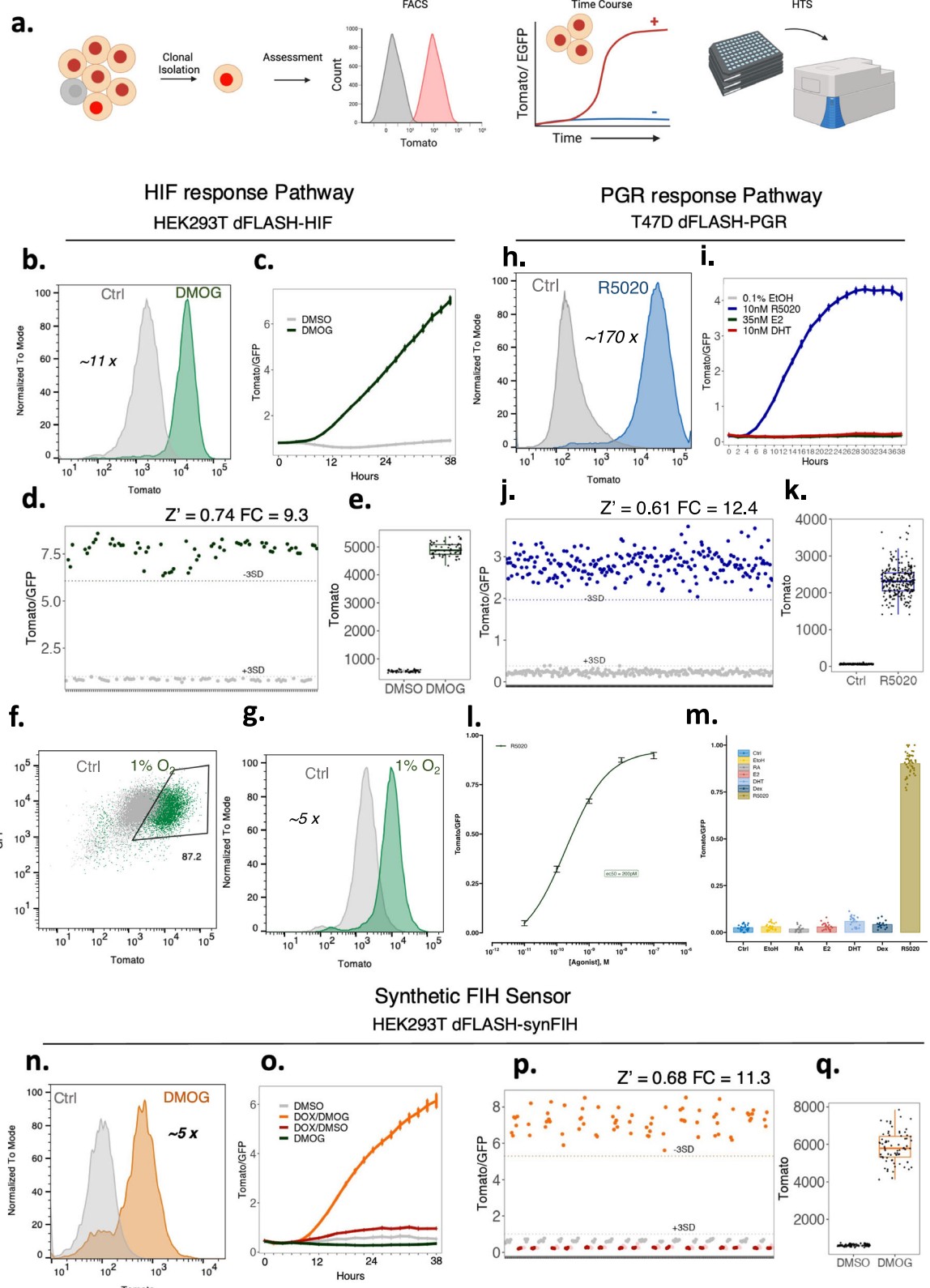

**HIF response Pathway**
HEK293T dFLASH-HIF

**PGR response Pathway**
T47D dFLASH-PGR

**Synthetic FIH Sensor**
HEK293T dFLASH-synFIH

$EC_{50} \sim 200$ pM, in agreement with orthogonal methods[34] (Fig. 3l, Supplementary Fig. 5f). This suggests that PRE-mcdFLASH responds sensitively and selectively to the PGR selective agonist R5020, and therefore has the potential for high-content screening for modulators of PGR activity. As such, we term this line mcdFLASH-PGR from herein, for its specific ability to report on PGR activity at physiological steroid concentrations.

For temporal HCI of populations (Figs. 2, 3), cells were imaged every 2 h, however this did not inherently provide single-cell temporal dynamics of transcriptional responses. Using clonally derived mcdFLASH-PGR or mcdFLASH-HIF lines, we also imaged transcriptional responses to R5020 or DMOG, respectively, every 15 min (Supplementary Movie 1 and 2). High temporal resolution imaging has the potential to monitor transcriptional dynamics in single cells, facilitated

**Fig. 3 | Derivation of robust, screen-ready, dFLASH clonal lines. a** Schematic for derivation and assessment of robustness in clonal lines, created in BioRender. Peet, D. (2025) https://BioRender.com/y10l388, of (**b**–**e**) HEK239T dFLASH-HIF (mcdFLASH-HIF), (**h**–**k**) T47D dFLASH-PGR (mcdFLASH-PGR) and (**n**–**q**) HEK293T dFLASH-synFIH (mcdFLASH-synFIH), analysed by flow cytometry, temporal HCI over 38 h and inter-plate robustness by mock multi-plate high throughput screening with HCI. **b**–**e** mcdFLASH-HIF was (**b**) treated with DMOG for 48 h and assessed for Tomato induction by flow cytometry relative to vehicle controls with fold change between populations stated and (**c**) treated with vehicle or 0.5 mM DMOG and imaged every 2 h for 38 h by HCI (mean ± sem, n = 8 biological replicates per group). **d, e** mcdFLASH-HIF was treated for 48 h with 1 mM DMOG or vehicle (6 biological replicates/plate, n = 10 plates) by HCI in a high throughput screening setting (HTS-HCI) for (**d**) normalised dFLASH expression and (**e**) Tomato MFI alone. **f, g** mcdFLASH-HIF was assessed by flow cytometry after a 24 h hypoxic (1% $O_2$) treatment, followed by a 4 h normoxic recovery period for (**f**) EGFP and Tomato and (**g**) Tomato alone, relative to a normoxic population (ctrl). **h**–**k** T47D mcdFLASH-PGR was (**h**) assessed after 48 h of treatment with 100 nM R5020 by flow cytometry for Tomato induction and (**i**) treated with 10 nM R5020, 35 nM E2, 10 nM DHT and vehicle then imaged every 2 h for 38 h by temporal HCI for normalised dFLASH expression (mean ± sem, n = 8 biological replicates per group). **j, k** T47D mcdFLASH-PGR was assessed by HTS-HCI at 48 h (48 biological replicates/plate, n = 5 plates) for (**j**) dFLASH normalised expression and (**k**) Tomato MFI alone. **l** T47D dFLASH-PGR cells were treated with increasing concentrations of R5020 (0.01-100 nM, n = 8 biological replicates per group, presented as mean ± sem) and imaged at 48 h to determine sensitivity to R5020. **m** Comparison of induction of the T47D mcdFLASH-PGR line to different steroids (10 nM R5020, 35 nM E2, 10 nM DHT, 10 nM Dex, 10 nM RA) by HCI after 48 h of treatment. **m** Are the mean ± sem of normalised Tomato/GFP (within each experiment) from n = 3 independent experiments (n = 8 biological replicates per group), except Dex and RA (n = 2 independent experiments (n = 8 biological replicates per group)). **n** HEK293T dFLASH-synFIH was assessed, with 200 ng/mL Dox +/- 1 mM DMOG by flow cytometry for Tomato induction. **o** mcdFLASH-synFIH was treated with 100 ng/mL Dox, 1 mM DMOG and relevant vehicle controls (0.1% water, 0.1% DMSO) and assessed for reporter induction by temporal HCI (mean ± sem, n = 4 biological replicates per group). **p, q** mcdFLASH-synFIH cells were treated with 200 ng/mL Dox (grey), 1 mM DMOG (red), vehicle (pink) or Dox and DMOG (orange) and assessed by HTS-HCI after 48 h (n = 8 biological replicates/plate, n = 3 plates) for (**p**) normalised dFLASH expression or (**q**) Tomato MFI induction between Dox only and Dox and DMOG treated populations. For (**d, j, p**) dashed lines represent 3 SD from relevant vehicle (+3 SD) or requisite ligand treated population (-3SD). Fold change for flow cytometry and HTS-HCI (FC) is displayed. Z' was calculated from all HTS-HCI analysed plates. Z' for all plates analysed was >0.5. **e, k, q** boxplots are presented as whiskers – 1.5 x IQR, box - 25th/75th percentile and median as the solid line. Source data are provided as a Source Data file.

by the dual fluorescent nature of dFLASH. Taken together this indicates that clonal lines display improved signal to noise and assay consistency, possibly enabling high content screening experiments.

Typically, high-content screening experiments require high in-plate and across plate consistency, therefore we evaluated mcdFLASH lines (HIF-1α, PGR, FIH) across multiple plates and replicates. System robustness was quantified with the Z' metric[35] accounting for fold induction and variability between minimal and maximal dFLASH outputs. Signal induced mnucTomato fluorescence across replicates from independent plates was highly consistent (Z' 0.61-0.74) and robust (9.3-12.4-fold, Fig. 3d, j, p) with the signal induced changes in activity for mcdFLASH-HIF and mcdFLASH-FIH driven by increased mnucTomato, (Fig. 3e, q). Despite the changes previously observed in nucEGFP in T47D mcdFLASH-PGR cells, these still provided equivalent reporter response to the other systems (Fig. 3j, k). As a result, mcdFLASH cell lines represent excellent high-throughput screening systems routinely achieving Z' scores >0.5. Importantly, the induction of the mcdFLASH lines (HEK293T and HepG2 mcdFLASH-HIF, T47D mcdFLASH-PGR and HEK293T mcdFLASH-SynFIH) remained stable over extended passaging (months), enabling protracted large screening applications.

## dFLASH-HIF CRISPRi-perturbations of the HIF pathway

The robust signal window and high Z' score of the mcdFLASH-HIF cell line, coupled with facile analysis by flow cytometry and HCI, suggested that the reporter system could be amenable to functional genomic screening. To investigate this, we utilised the recently developed CRISPRoffv2.1 system[36] to stably repress expression of *VHL*, which mediates post-translational downregulation of the HIF-1α pathway[37,38]. We generated stable mcdFLASH-HIF cells expressing a guide targeting the *VHL* promoter and subsequently introduced CRISPRoffv2.1 from either a lentivirus driven EF1α or SFFV promoter (Fig. 4a, b). Cells were then analysed by flow cytometry 5- or 10-days post selection to determine if induction of mcdFLASH-HIF reporter was modulated by *VHL* knockdown under normoxic conditions (Supplementary Fig. 7, Fig. 4c, d). As expected, mcdFLASH-HIF/sgVHL cells expressing CRISPRoffv2.1 from either promoter induced the mcdFLASH-HIF reporter ~30% by 5 days and the majority of cells (~60%) by 10 days, as compared to cells expressing a HIF pathway independent *PGR* sgRNA (Supplementary Fig. 7e), or no sgRNA (Fig. 4c, d, Supplementary Fig. 7b). This demonstrates that mcdFLASH-HIF is responsive to CRISPRi/off perturbations of key regulators of the HIF pathway and therefore illustrates its potential to act as a readout of CRISPR screens at-scale in a larger format, including genome-wide screens.

## High performance inducible whole genome pooled CRISPR screens with dFLASH-HRE

In order to investigate the performance of the dFLASH-HIF system in pooled CRISPR KO screens assessing the effect of essential genes on the HIF pathway, we engineered dox-inducible Cas9 expressing cell lines that can be established with blasticidin selection, with Cas9 expression monitored and enriched using tagBFP. We established U2OS and HEK293T mcdFLASH-HIF/iCas9-tagBFP cell lines for inducible knockout experiments, noting the U2OS reporter line displayed a 436-fold mean induction with DMOG treatment (Fig. 5a). Having demonstrated that knockdown of *VHL* was sufficient to activate the dFLASH-HRE system, we turned to short term ablation, by sgRNA nucleofection, of *HIF1A* and *VHL* to confirm that both loss of function and gain of function knockout screens could be performed. Following 3 days of dox treatment and 1–2 days of DMOG treatment, we observed near complete loss of activation in *HIF1A* sgRNA expressing cells or strong homogenous activation in *VHL* sgRNA expressing cells (Fig. 5a). Taken together, this indicates that short term knockout of key pathway regulators is sufficient to completely shift reporter activity. Next, we explored the performance of the dFLASH system in pooled screens by continuing with the HIF pathway and performing two whole genome KO screens in U2OS and HEK293T mcdFLASH-HIF/iCas9-tagBFP lines using the Brunello KO library containing ~77,000 sgRNAs targeting ~19,000 genes (Fig. 5b). One key advantage of the dFLASH system is the presence of a genome loci control *EGFP* gene driven by a constitutive promoter, allowing exclusion of genes that control general and common mechanisms of transcription or other cellular processes. As such, we introduced the pooled Brunello library and induced knockout with a short-term regime (5 days) of Dox treatment and subsequent pathway activation with DMOG, sorting low Tomato expressing cells that maintained EGFP expression. We identified hundreds (MAGeck - HEK = 522, U2OS = 501 or DEseq2 - HEK = 322, U2OS = 441) of previously unreported and known regulators of the HIF pathway enriched in loss-of-function pools (Fig. 5c, d). In order to benchmark the performance of the dFLASH system to previous HIF pathway CRISPR screens, we compared an analogous loss of function screen of the HIF pathway using a 3xHRE-mCherry[ODD] (mCherry fused to an oxygen dependent degradation domain) driven reporter[9]. While there are substantial differences between these screening approaches (different libraries and selection strategies) dFLASH screening in U2OS or HEK293T cells substantially outperformed the previous screen by identifying 522 and 501 significantly enriched hits in the low activity pools versus 36 hits in the 3xHRE- mCherry[ODD] screen, indicating that

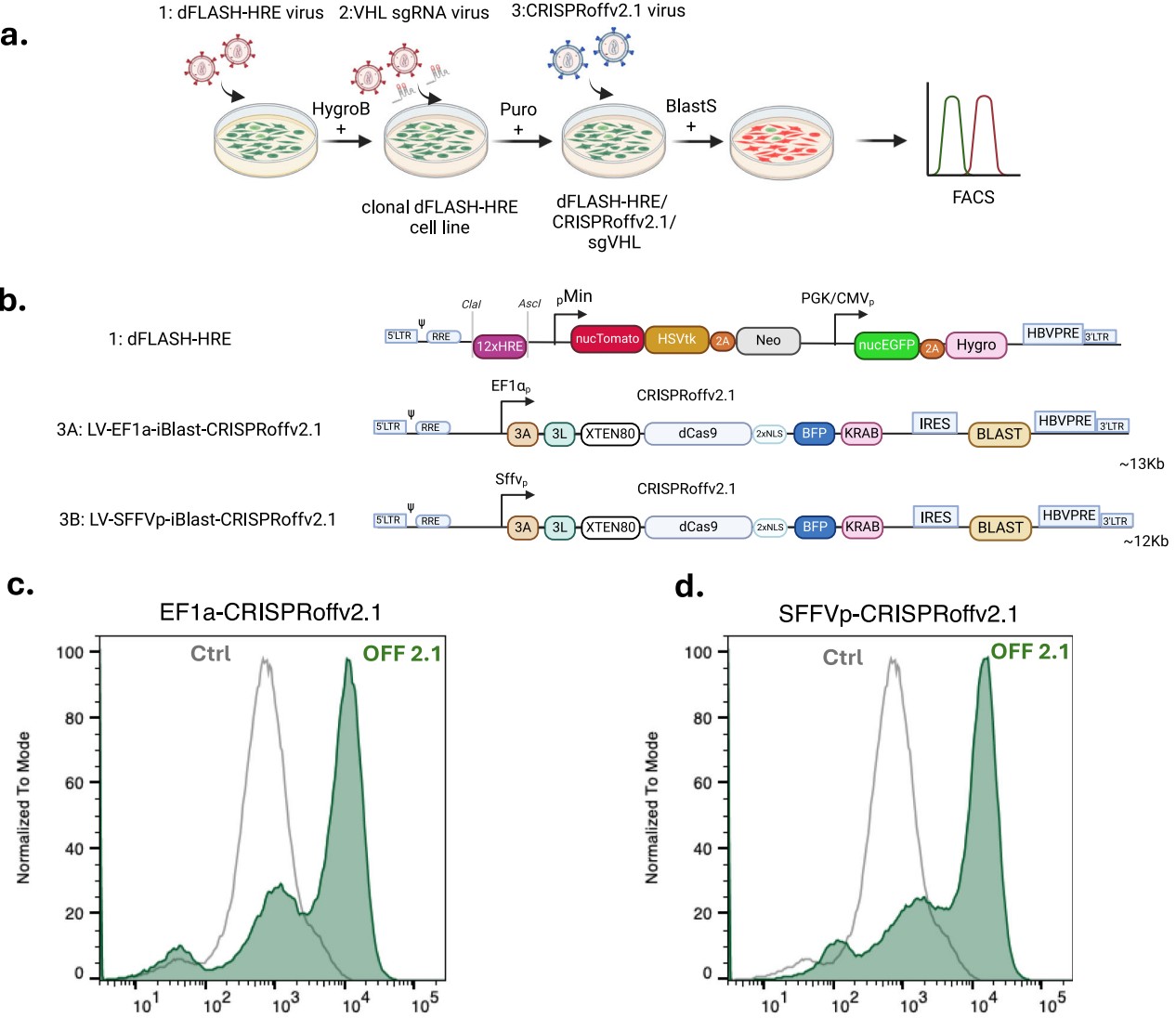

**Fig. 4 | Near homogenous activation of mcdFLASH-HIF by CRISPRoff knock-down of VHL. a** Clonal (1) mcdFLASH-HIF lines derived post-hygromycin (HygroB) selection were transduced first with the (2) sgRNA vector targeting the *VHL* transcriptional start site, followed by puromycin selection (Puro). This pool was subsequently transduced with the (3) CRISPRoffv2.1 virus and selected with blasticidin (BlastS) prior to flow cytometry (on day 5 and 10 post BlastS selection). **b** The (1) dFLASH vector with the HRE enhancer was transduced, as were 2 variants of the CRISPRoffv2.1 vector with either (**3A**) EF1α promoter or (**3B**) SFFV promoter driving the dCas9 expression cassette. 3A denotes Dnmt3A, 3L denotes Dnmt3L, XTEN80 denotes an 80 amino acid linker sequence for improved protein half-life. **a**, **b** created in BioRender. Peet, D. (2025) https://BioRender.com/l71s669. **c**, **d** Flow cytometry for dFLASH-HIF induction in response to the CRISPRoffv2.1 VHL knockdown relative to parental line (ctrl) with (**c**) EF1α or (**d**) SFFV expression constructs after 10 days of selection. Data are representative of *n* = 2 independent experiments.

the dFLASH system is well suited to pooled screening approaches (Fig. 5c). In addition, we found strong enrichment of the known obligate pathway regulators *HIF1A (HEK293T; p = 2.65 × 10^{-15}, U2OS; p = 8.17 ×10^{-14})* and *ARNT (HEK293T; p = 1.1 × 10^7, U2OS; p = 1.74 × 10^{-32})* in both screens, indicating that bonefide HIF regulators are likely to be identified using such screening approaches (Fig. 5d).

## dFLASH facilitates bimodal screening for small molecule discovery

Manipulation of the HIF pathway is an attractive target in several disease states, such as in chronic anaemia[39] and ischaemic disease[40], where its promotion of cell adaption and survival during limiting oxygen is desired. Conversely, within certain cancer subtypes[41,42] HIF signalling is detrimental and promotes tumorigenesis. Therapeutic agents for activation of HIF-α signalling through targeting HIF-α regulators was initially discovered using in vitro assays. However, clinically effective inhibitors of HIF-1α signalling are yet to be discovered[43]. The

biological roles for HIF-1α and closely related isoform HIF-2α, which share the same canonical control pathway, can be disparate or opposing in different disease contexts, requiring isoform selectivity for therapeutic intervention[44]. To validate that HIF-1α is the sole isoform regulating mcdFLASH-HIF in HEK293T cells[45], tandem HA-3xFLAG epitope tags were knocked in to the endogenous HIF-1α and HIF-2α C-termini allowing direct comparison by immunoblot[46]. This confirmed that HIF-1α is the predominant isoform in HEK293T cells (Supplementary Fig. 6a). Furthermore, there was no change in DMOG induced mnucTomato expression in HEK293T mcdFLASH-HIF cells when co-treated for up to 72 h with the selective HIF-2α inhibitor PT-2385 (Supplementary Fig. 6b), consistent with the minimal detection of HIF-2α via immunoblot. This confirmed that our HEK293T dFLASH-HIF cell line specifically reports on HIF-1 activity and not HIF-2, indicating that it may be useful for identification of drugs targeting the HIF-1α pathway. Furthermore, dFLASH-HIF was able to respond to the PHD-specific inhibitor FG-4592 (Supplementary Fig. 6c), providing

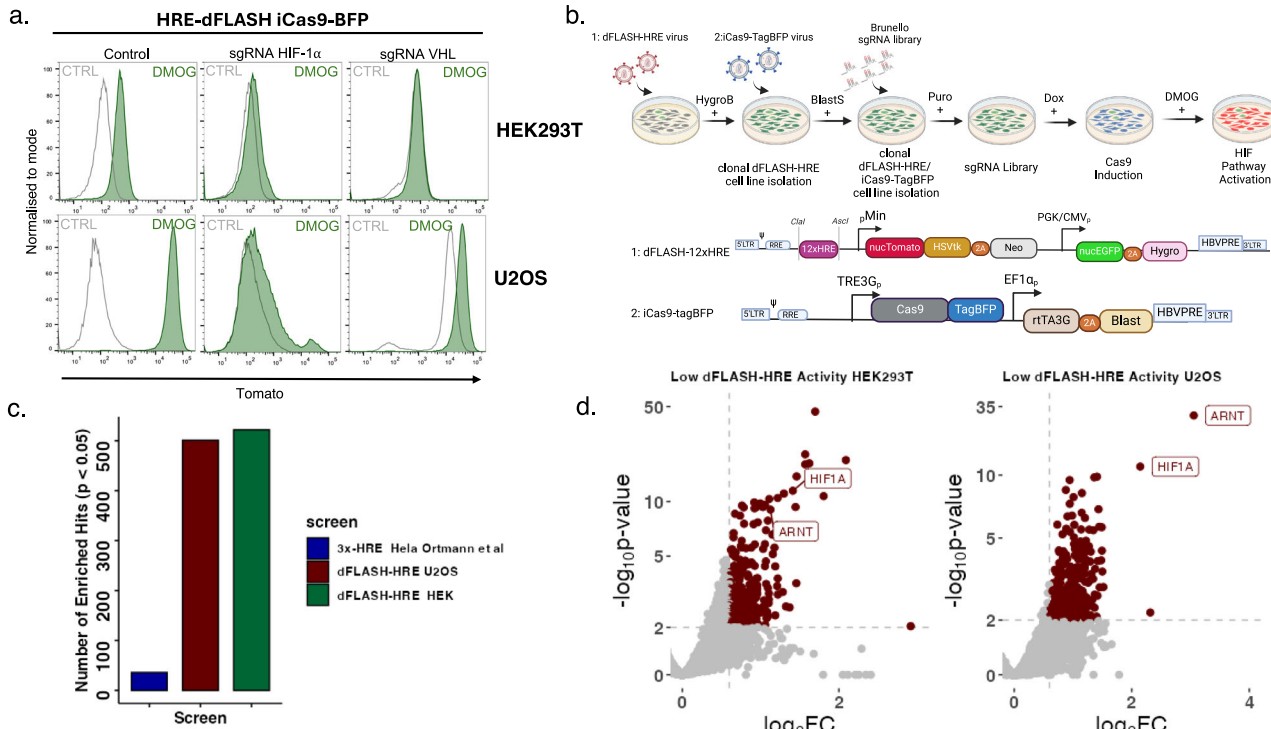

**Fig. 5 | Inducible Cas9 whole genome CRISPR KO screens using dFLASH-HIF.**
**a** HEK293T (upper panel) or U2OS (lower panel) mcdFLASH-HIF/iCas9-BFP cells
were used to validate CRISPR KO loss of function screening by short term (3 days)
inducible (dox) knockout of HIF-1α or gain of function with knockout of VHL. Data
is representative of *n* = 2 independent experiments. **b** Schematic of the whole
genome KO screening strategy. Briefly, clonal dFLASH-HRE reporter lines were
transduced with a dox inducible Cas9-TagBFP virus and selected with blasticidin
(BlastS) and for homogeneous Cas9-BFP induction. The Brunello whole genome KO
library, ~77,000 guide RNAs targeting 19,114 genes, was then introduced and cells
selected with puromycin (Puro) prior to a short-term (5 days) Cas9 induction with
dox and FACS isolation of low activity pools. Created in BioRender. Peet, D. (2025)

https://BioRender.com/m13q226 **c** dFLASH-HIF benchmarking against another
published pooled HIF CRISPR screen. Loss of function CRISPR screen performed
with Hela 3xHRE-mCherry[ODD] (Ortmann et al. 2021) vs HEK293T or U2OS dFLASH-
HIF CRISPR screens. The number of enriched MAGeck CRISPR hits (*p* < 0.05, one-
tailed) were quantified and plotted. **d** Volcano plots of dFLASH-HRE CRISPR screens
in HEK293T (left panel) or U2OS (right panel) demonstrating the strong selection of
obligate regulators HIF-1α and ARNT. Differentially regulated genes from DEseq2
(generalised linear model with negative binomial distribution) with a *p*-value <
0.001 (grey dotted horizontal line, two-sided Wald test) and >0.6 LFC (grey dotted
vertical line) are marked as red dots. Data presented is from *n* = 3 independent
CRISPR screens. Source data are provided as a Source Data file.

further evidence towards its application for the identification of drugs
that target the HIF-1α pathway, and its direct regulators such as
the PHDs.

dFLASH-HIF facilitates multiple measurements across different
treatment regimens and time points, enabling capture of periodic
potentiated and attenuated HIF signalling during a single experiment.
Having validated the robust, consistent nature of mcdFLASH-HIF
above, we exploited its temporal responsiveness for small molecule
discovery of activators or inhibitors of HIF-1α signalling in a single,
bimodal screening protocol. To test this bimodal design, we utilised a
natural product library of 1595 compounds containing structures that
were unlikely to have been screened against HIF-1α prior.

We assayed compounds for normoxic activation of the
mcdFLASH-HIF reporter, followed by assessing compound inhibition
after DMOG stimulation (Fig. 6a). This allowed for both pathway acti-
vators and inhibitors to be identified in a single experimental para-
digm. We classified hits as those that displayed a Tomato/EGFP z-score
≥ 2 (activator) or ≤ 2 (inhibitor) without changes in EGFP (see
methods, Supplementary Fig. 8a–k). We initially screened compounds
at 36 h after cell seeding or DMOG addition, identifying 25 (Fig. 6b,
1.4% hit rate) putative activators and 69 inhibitors (Fig. 6c, 4.2% hit
rate). To explore the consistency of this screening approach and
identify bonafide hits, we replicated the compound screen whilst
shortening compound incubations to 24 h post cell seeding or DMOG
stimulation to reduce any toxicity related effects. In replicate screens
we identified 8 (Fig. 6d, 0.5% hit rate) putative activators and 81

inhibitors (Fig. 6e, 5.07% hit rate). Consistency of the compound
activities between the two screens was assessed by Pearson correla-
tions (Supplementary Fig. 8m and n, activators - *R* = 0.79, *p* < 2.2 × 10⁻¹⁶,
inhibitors - *R* = 0.62, *p* < 2.2 × 10⁻¹⁶), indicating a high degree or repro-
ducibility between the screens. Together, we identified 3 overlapping
putative activators (Fig. 6b, d, 0.18% combined hit rate) and 36 over-
lapping putative inhibitors (Fig. 6c, e, 2.25% combined hit rate) in line
with reports of analogous agonist and antagonist hit rates[47].

**dFLASH identified previously unreported and known com-
pounds that alter HIF TF activity**
From the list of putative hits generated from the bimodal screening,
and review of the performance between both time points, we selected
11 inhibitors and 18 activators for reassessment using a three point
concentration curve (Fig. 7, Supplementary Figs. 9, 10). This led to
identification of RQ500235 and RQ200674 (Fig. 7a, d) as previously
unreported HIF-1α inhibiting or stabilising compounds, respectively.
RQ200674 increased reporter activity 2-fold in repeat assays (Fig. 7d),
outperforming the other assessed activators (Supplementary Fig. 9a)
and stabilised endogenously tagged HIF-1α at normoxia in HEK293T
cells (Supplementary Fig. 9b). Mechanistically, RQ200674 had weak
iron chelation activity in an in vitro iron chelation assay (Fig. 7e),
suggesting it intersects with the HIF-1α pathway by sequestering iron,
similar to other reported HIF stabilisers. In the inhibitor compound
screens, several compounds demonstrated consistent reporter
downregulation (Fig. 7a, Supplementary Fig. 10a). Celastarol

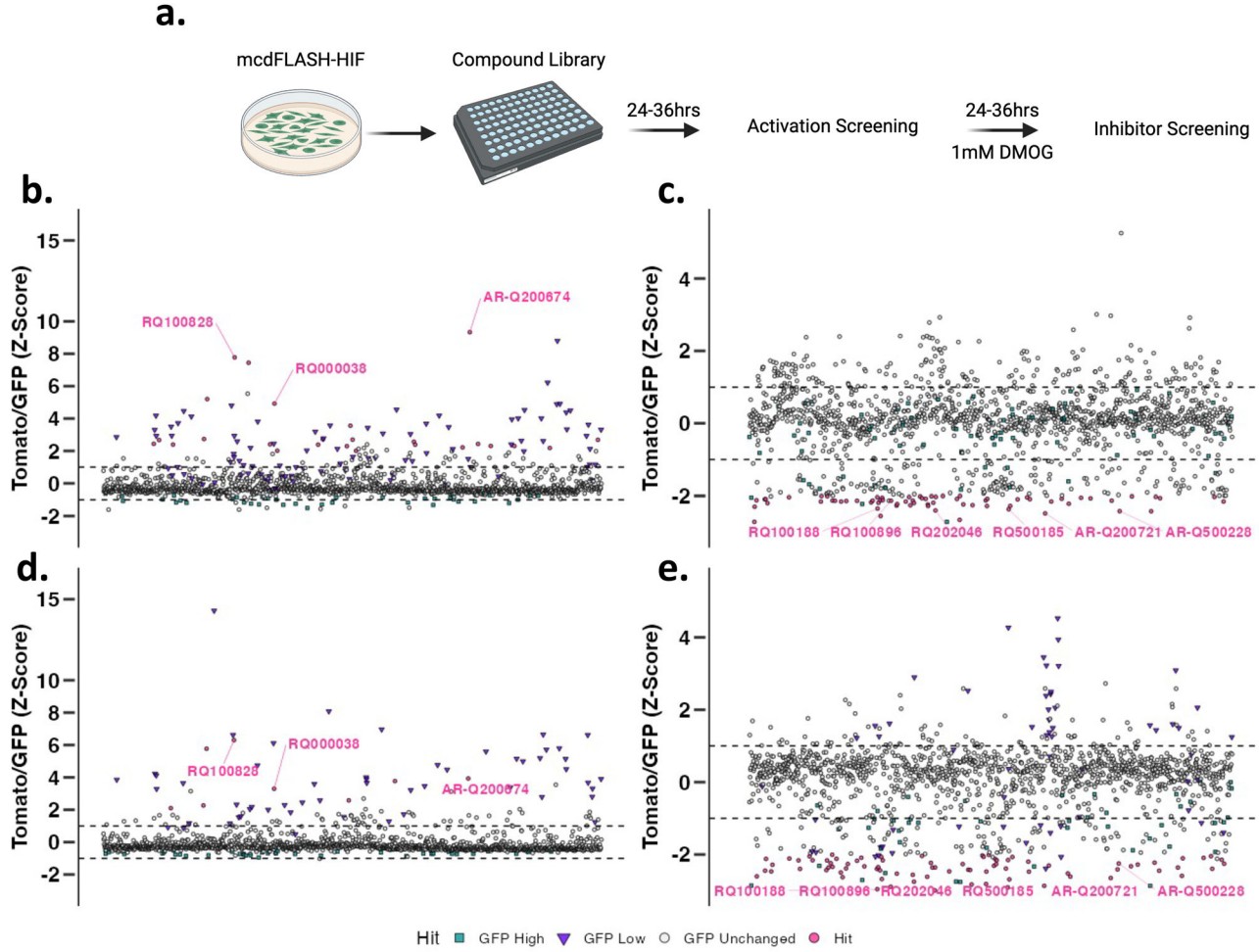

**Fig. 6 | Bimodal small molecule screening of the HIF signalling pathway using dFLASH-HIF to identify positive and negative regulators. a** HEK293T mcdFLASH-HIF cells were treated with a 1595 compound library and incubated for 36 (**b**) or 24 (**d**) hours prior to the first round of HCI. After the first round of screening, the compound wells were then treated with 1 mM DMOG for 36 (**c**) or 24 (**e**) hours prior to the second round of HCI. Created in BioRender. Peet, D. (2025) https://BioRender.com/c07j152. **b, d** Compounds that increase Tomato/EGFP > 2 SD or (**c, e**) compounds that decreased Tomato/EGFP > 2 SD are counted as hits and coloured in pink circles. All compounds that changed EGFP > ± 2 SD were excluded (purple triangles or green squares). Tomato and EGFP values are shown in Supp. Fig. 8. Dashed lines indicate 1 SD thresholds. Normalised dFLASH output (Z scoring) for all compound-treated wells is shown. Singlet compound wells were screened per screening replicate at each time point. Vehicle (*n* = 80) and positive controls (1 mM DMOG, *n* = 80 activator screening, *n* = 160 for inhibitor screening) were included on each screening plate. Source data are provided as a Source Data file.

(RQ000155) and Flavokawain B (RQ100976) downregulated the reporter at several concentrations (Supplementary Fig. 10b, c). Celastarol is a previously reported HIF-1α inhibitor[48–50] and Flavokawain B is a member of the chalcone family which has previously exhibited anti-HIF-1α activity[51], thus validating our screening design. RQ500235 was identified as a previously unreported HIF-1 inhibitor by mcdFLASH-HIF screening. Dose dependent inhibition of mcdFLASH-HIF (Fig. 7a) correlated with a dose-dependent decrease in protein expression by immunoblot (Fig. 7b). We observed a significant (*p* = 0.0139) down-regulation of *HIF1A* transcript levels (Fig. 7c) and were unable to rescue HIF-1α protein loss with proteasomal inhibition (Supplementary Fig. 10d), indicating RQ500235 was decreasing HIF-1α at the mRNA level. More broadly however, the identification of these compounds by mcdFLASH-HIF in the bimodal screen set up demonstrated successful application of the dFLASH platform to small molecule discovery efforts for both gain and loss of TF function.

## Discussion

We designed and optimised dFLASH to offer a versatile, robust, live-cell reporting platform that is applicable across TF families and allows for facile high-throughput applications. We validated dFLASH against three independent signal-responsive TFs, two with endogenous signalling pathways (dFLASH-PRE for Progesterone Receptor; dFLASH-HRE for hypoxia induced transcription factors) and a synthetic system for hybrid protein transcriptional regulators (dFLASH-Gal4RE). Each dFLASH construct produced robustly detected reporter activity by temporal high-content imaging and FACS after signal stimulation for its responsive TF (Figs. 2, 3). The use of previously validated enhancer elements for HIF[25] and synthetic Gal4 DNA binding domains[23,29] demonstrated that dFLASH can be adapted towards both endogenous and synthetic pathways displaying highly agonist/activator-specific responses, indicating utility in dissecting and targeting distinct molecular pathways. The distinct pathways targeted by the mcdFLASH lines produced highly consistent (Z′ = 0.62-0.74) signal induced Tomato induction measured by high content imaging, suggesting dFLASH is highly suited to arrayed high-throughput screening (Fig. 3). In addition, mcdFLASH lines also displayed homogenous signal induced reporter induction by flow cytometry indicating that pooled high content screening is also possible.

Indeed, reporter systems like dFLASH have been increasingly applied to functional genomic screens which target specific transcriptional pathways[9,17,52,53]. However, none of these incorporate an

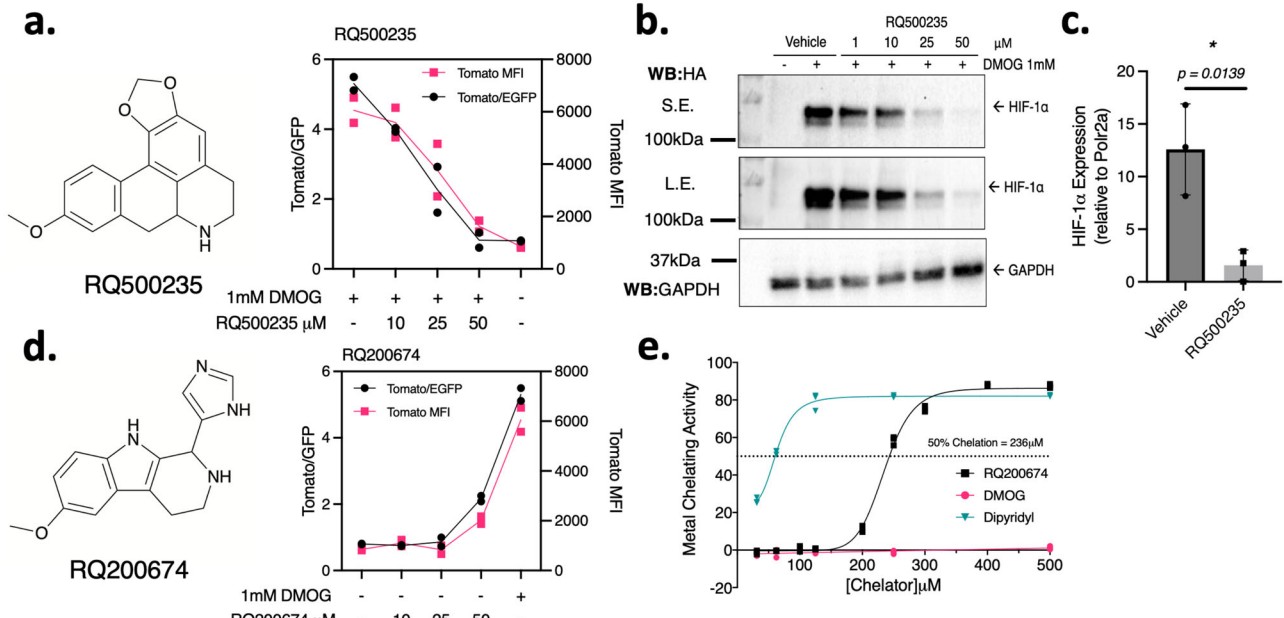

**Fig. 7 | Investigating the mechanisms of HIF-1α regulation by dFLASH-HIF inhibitor hit RQ500235 and activator hit RQ200674. a, b** Inhibitor RQ500235 identified from the bimodal screen (**a**) represses DMOG induced Tomato in dFLASH-HIF cells in a dose dependent manner ($n = 2$ biological replicates) and (**b**) decreases expression of HIF-1α protein as assessed by immunoblot of whole cell extracts from endogenous HA-Flag tagged HIF-1α in HEK293T cells. S.E. short exposure; L.E. long exposure. Representative of 3 independent replicates. **c** RT-qPCR shows HIF-1α transcript is significantly decreased in HEK293T cells treated for 6 h with RQ500235 ($n = 3$ biological replicates, *$p = 0.0139$ assessed by a two-tailed $t$-test with Welch's correction, presented as mean relative expression ± SD). **d** Activator RQ200674 identified from the bimodal screen recapitulated activation of dFLASH-HIF at 50 μM in HEK293T cells ($n = 2$ biological replicates). **e** In vitro iron chelation assay of RQ200674 which displays weak chelating activity at 236 μM from line of best fit ($n = 3$ independent experiments) compared to positive control iron chelator and HIF-1α activator, dipyridyl. Source data are provided as a Source Data file.

internal control at the same genomic loci, making filtering of non-pathway specific hits a challenge. CRISPRoff mediated downregulation of the core HIF protein regulator, *VHL*, produced distinct tomato expressing dFLASH-HIF cell pools (Fig. 4), demonstrating genetic perturbations of endogenous TF signalling pathways. Indeed, we also show that the dFLASH system is well suited to pooled whole genome CRISPR KO screens, substantially outperforming a previous similar reporter-based HIF screen (Fig. 5). This suggests that the improved utility of a loci specific internal control which dFLASH provides would be advantageous in other transcriptional genomic screens (see Supplementary Data 1 for comparison of dFLASH with analogous platforms)[9,17,52,54].

Using the HIF-1α specific reporter line, HEK293T mcdFLASH-HIF, the application of arrayed high-content drug screening was also demonstrated. This approach was successful in discovering a previously unreported small molecule activator and inhibitor of the HIF pathway, as well as previously identified inhibitory compounds. This ratified dFLASH as a reporter platform for arrayed-based screening and demonstrates the utility of the linked nucEGFP control for rapid hit bracketing (Fig. 6). The previously unreported HIF-1α inhibitor RQ500235 was shown to downregulate *HIF1A* transcript levels, like another HIF-1α inhibitor PX-478[55,56]. As PX-478 has demonstrated anti-cancer activity in several cell lines[56,57] and preserved β-cell function in diabetic models[55], a future similar role may exist for an optimised analogue of RQ500235.

The dFLASH system is characterised by distinct advantages which may enable more precise dissection of molecular pathways. The ability to control for cell-to-cell fluctuations and to decouple generalised or off-target effects on reporter function, may aid the precision necessary for large drug library or genome-wide screening applications[58]. In addition, dFLASH, unlike many other high-throughput platforms, can be used to screen genetic or drug perturbations of temporal transcriptional dynamics, or, as used here, at multiple time points.

Furthermore, these results demonstrate that dFLASH is ideally suited to array-based functional genomics approaches[59] allowing for multi-plexing with other phenotypic or molecular outputs[2,60–62]. Importantly, dFLASH provides distinct advantages over bioluminescent based reporter systems which cannot be subject to temporal live single cell imaging or FACS based analysis. Furthermore, the dual fluorescent protein nuclear segmentation and image quantitation enables single cell reporter normalisation from the same genomic locus, providing a real-time, relative and temporal assessment of transcriptional responses, which has not been previously achieved by other systems. The extensive optimisation of dFLASH design (Supplementary Fig. 1), together with application to two endogenous transcription factor pathways and synthetic transcription factors, all benchmarked using the Z' screening metric, highlights the distinct advantage of dFLASH over currently available systems and its versatility to robustly report on endogenous and synthetic TFs by high content imaging or flow cytometry.

The dFLASH approach does have some limitations, however. The fluorescent nature of dFLASH limits the chemical space by which it can screen due to interference from auto-fluorescent compounds. In addition, we acknowledge that fluorescent proteins require $O_2$ for their activity and this limits the use of mnucTomato as a direct readout of hypoxia (in the absence of a normoxic recovery period)[63]. Additionally, while the backbone design has been optimised for robust activation of a variety of transcriptional response pathways, the mechanistic underpinning of this is unclear and could be further improved, providing insights into the sequence and architectural determinants of enhancer activation in chromatin. In addition to the strong effect of the dFLASH downstream promoter on upstream enhancer activity (Supplementary Fig. 1) it is clear that either the distance between contiguous promoter/enhancer or the sequence composition of the linker has a functional consequence on enhancer induction. Integration of these design principles with high throughput assessments of

transcriptional reporter design may further improve the implementation of robust pathway specific reporters, especially in primary cells[28].

The incorporation of robust native circuits, such as those described here (Hypoxia or Progesterone), has the potential to allow the manipulation or integration of these pathways into synthetic biology circuitry for biotherapeutics. In these cases, it is critical that robust signal to noise is achieved for these circuits to effectively function in biological systems. Further, the use of a synthetic approach to 'sense' FIH enzymatic activity through the HIFCAD:p300/CBP interaction opens up the possibility that other enzymatic pathways that lack an effective in vivo activity assay may also be adapted. We also envisage that dFLASH could be adapted to 2-hybrid based screens as a complement to other protein-protein interaction approaches.

The ability to temporally track TF regulated reporters in populations and at the single-cell level enables dFLASH to be used to understand dynamics of transcriptional responses, as has previously been used to dissect mechanisms of synthetic transcriptional repression[7,8] or to understand notch ligand induced synthetic transcriptional dynamics[64]. For instance, synthetic reporter circuits have been used to delineate how diverse notch ligands induce different signalling dynamics[64]. The large dynamic range of the dFLASH-PGR and HIF reporter lines in conjunction with the high proportion of cells induced in polyclonal pools (Fig. 2) also suggests dFLASH is a candidate system for forward activity-based enhancer screening. These approaches have been applied to dissect enhancer activity or disease variants with other similar systems such as lentiviral-compatible Massively Parallel Reporter Assays (LentiMPRA)[65,66]. However, the use of the internal control normalisation provided by dFLASH may be useful in separating chromosomal from enhancer driven effects in forward enhancer screens. In addition, the dFLASH-PGR reporter described here is ideally suited for high throughput chemical and genetic perturbation screens to identify non-hormonal regulators of the progesterone signalling pathway.

Together, we demonstrate that dFLASH displays robust activity in both pooled and arrayed formats and therefore provides a flexible platform for a variety of investigations, including but not limited to, pooled genomic screening and arrayed small molecule drug discovery. dFLASH can be used to sense both endogenous and synthetic transcription factor activity and thus represents a versatile, stable, live-cell reporter system for a broad range of applications.

## Methods
### Ethical statement
All animal experiments were approved by The University of Adelaide Faculty of Health and Medical Sciences Animal Ethics Committee and were performed in accordance with the Australian Code of Practice for the Care and Use of Animals for Scientific Purposes; ethics approval number M-2023-074.

### Animals
Twenty-one day old female CBA CaH x C57BL/6 F1 (CBAF1) mice used in this study were attained from the University of Adelaide Laboratory Animal Services. Water and chow was provided ad libitum and mice were housed in 12 h light / 12 h dark conditions under constant temperature and humidity.

### Plasmid Construction.
All new Plasmids constructed as part of this work are available at Addgene https://www.addgene.org/David_Bersten/. Briefly, cDNAs were amplified using Phusion polymerase (NEB) and assembled into ClaI/NheI digested pLV410 backbone by Gibson assembly[32]. Sequence verified LV-REPORT plasmid sequences and constructs are listed in Supplementary Table 1. Briefly, the plasmids contained an upstream multiple cloning site followed by a minimal promoter optimised for low background expression[18], then followed by a mnucTomato/HSVtk-2a-Neo reporter construct or other

variants. This was then followed by a constitutive promoter (EF1α, PGK or PGK/CMV) driving the expression of a hygromycinR cassette with or without a 2A linked d2nucEGFP (Supplementary Fig. 1C).

To improve the performance of our previously reported lentiviral inducible expression systems[67], the PGK promoter in Tet-On3G IRES Puro was replaced by digestion with MluI/NheI and insertion of either EF1α-Tet-On3G-2A-puro, EF1α-Tet-On3G-2A-BlastR or EF1α-Tet-On3G-2A-nucTomato using Phusion polymerase (NEB) amplified PCR products from existing plasmids. Plasmids were cloned by Gibson isothermal assembly and propagated in DB3.1 cells (Invitrogen). We also generated a series of constitutive lentiviral plasmids as part of this work; pLV-EgI-BlastR (EF1α-Gateway-IRES-BlastR, Addgene #207176), pLV-EgI-ZeoR (EF1α-Gateway-IRES-ZeoR, Addgene #207178), pLV-EgI-HygroR (EF1α-Gateway-IRES-HygroR, Addgene #207177), pLV-SFFVp-gI-BlastR (SFFVp-Gateway-IRES-BlastR, Addgene #207179) and pLV-SV40p-gI-BlastR (SV40p-Gateway-IRES-BlastR, Addgene #207182). These plasmids were constructed by isothermal assembly of G-Blocks (IDT DNA) or PCR fragments, propagated in ccbD competent cells, sequence verified and deposited on Addgene (Supplementary Table 1).

The Lentiviral backbone expression construct pLV-TET2BLAST-GtwyA was then used to insert expression constructs cloned into pENTR1a by LR Clonase II enzyme recombination (Cat#11791020, Thermo). GAL4DBD-HIFCAD (727-826aa) and the GAL4DBD[29] were cloned into pENTR1a by ScaI/EcoRV or KpnI/EcoRI respectively. The miniVPR sequence[30] was cloned into the pENTR1a-GAL4DBD construct at the EcoRI and NotI sites. The pENTR1a vectors were then Gateway cloned into the pLV-TET2PURO-GtwyA vector. pENTR1a-Cas9-TagBFP was cloned by isothermal assembly and sequence verified prior to transferring to pLV-TET2BLAST by LR Clonase II enzyme recombination. pENTR1a-CRISPRoffv2.1 was generated by inserting an EcoRI/NotI digested CRISPRoff2.1 (CRISPRoff-v2.1 was a gift from Luke Gilbert, Addgene #167981) into pENTR1a plasmid. pLV-SFFVp-CRISPRofv2.1-IRES-BLAST (Addgene #207181) and pLV-EF1α-CRISPRofv2.1-IRES-BLAST (Addgene #207180) were generated using pENTR1a by LR Clonase II enzyme recombination (Cat#11791020, Thermo). All Lentiviral plasmids were propagated in DH5α without any signs of recombination.

### Enhancer element cloning.
The 12x HRE enhancer from hypoxic response target genes (*PGK1*, *ENO1* and *LDHA*) was liberated from pUSTdS-HRE12-mCMV-lacZ[25] with XbaI/SpeI and cloned into AvrII digested pLV-REPORT plasmids. Progesterone responsive pLV-REPORT-PRECat was cloned by isothermal assembly of a G-Block (IDT DNA) containing enhancer elements from five PGR target gene enhancers (Zbtb16, Fkbp5, Slc17a11, Erfnb1, MT2)[32] into AscI/ClaI digested pLV-REPORT(PGK/CMV). Gal4 response elements (5xGRE) were synthesised (IDT DNA) with ClaI/AscI overhangs and cloned into Cla/AscI digested pLV-REPORT(PGK/CMV). Sequences are in Supplementary Data 2.

### Mammalian cell culture and ligand treatment.
HEK293T (ATCC CRL-3216), HEPG2 (ATCC HB-8065), or U2OS cell lines (verified by STR sequencing) were grown in Dulbecco's Modified Eagle Medium (DMEM high glucose, Thermo #11965092) + pH 7.5 HEPES (Gibco), 10% Foetal Bovine Serum (Corning 35-076-CV or Serana FBS-AU-015), 1% penicillin-streptomycin (Invitrogen) and 1% Glutamax (Gibco, #35050061). T47D (ATCC HTB-133) or BT474 (ATCC HTB-20) were grown in RPMI 1640 (ATCC modified) (Gibco, A1049101) with 10% Foetal Bovine Serum (Fisher Biotech FBS-AU-015) and 1% penicillin-streptomycin[68]. Cells were maintained at 37 °C and 5% $CO_2$. Clonal lines were isolated by either limiting dilution or FACS single cell isolation into 96 wells trays. Resultant monoclonal populations were evaluated by HCI or FACS and lines displaying homogenous robust induction were selected. Ligand treatments were done 24 h after seeding of cells in requisite plate or vessel. Standard concentrations and solvent,

unless specified otherwise, are 200 ng/mL Doxycycline (Sigma, D9891, $H_2O$), 0.5 mM or 1 mM DMOG (Cayman Scientific, 71210, DMSO), 100 nM R5020 (Perkin-Elmer, NLP004005MG, EtOH), 35 nM Estradiol (E2, Sigma E2758, EtOH), 10 nM all-trans retinoic acid (RA, Sigma #R2625), 10 nM Dihydrotestosterone (DHT, A8380 Sigma-Aldrich), 10 nM Dexamethasone (Dex, Sigma D4902), 10 μM PT-2385 (Abcam, ab235501, DMSO) and 50 μM FG-4592 (Cayman Chemical, 15294, DMSO). For hypoxic treatments, HEK293T-mcdFLASH-HIF cells were seeded in 6 well dishes at $2 \times 10^5$ cells per dish. Hypoxic incubations were completed in an Oxford Optronics HypoxyLab under humidified conditions at 1% $O_2$, 37 °C and 5% $CO_2$. Hypoxic-treated populations were incubated for 4 h under normoxic conditions prior to flow cytometry.

**Lentiviral production and stable cell line production.** Near confluent HEK293T cells were transfected with either psPAX2 (Addgene #12260) and pMD2.G (Addgene #12259) or pCMV-dR8.2 dvpr (Addgene #8455), pRSV-REV (Addgene; #12253) and pMD2.G along with the lentivector (described above) and PEI (1 μg/μl, polyethyleneimine) (Polysciences, USA), Lipofectamine 2000 (Thermo, #11668027), or Lipofectamine 3000 (Thermo, # L3000015) at a 3 μl:1 μg ratio with DNA. Media was changed 1-day post-transfection to complete media or Optimem (ThermoFisher; #31985088). Virus was harvested 1–2 days post-transfection, then filtered (0.45 μM (SART-16533-K) or 0.22 μM (SART-16532-K), Sartorius) before the target cell population was transduced at a MOI < 1. Cells were incubated with virus for 48 h prior to media being exchanged for antibiotic containing complete media. Standard antibiotic concentrations were 140 μg/mL Hygromycin B (Thermo, #10687010), 1 μg/mL Puromycin (Sigma; #P8833) or 10 μg/mL Blasticidin S (Sigma; #15205).

**Generation of CRISPR/Cas9 knockout or CRISPRoffv2.1 knockdown cell lines**
FIH CRISPR knockout guides and plasmids were obtained from Chen et al. (2017)[69]. These guides were transfected into HEK293T cells and with PEI at a 3 μg:1 μg ratio. FIH knockouts were selected via serial dilution and confirmed with PCR amplification, T7E1 assay and sanger sequencing coupled with CRISPR-ID[70]. The VHL or PGR sgRNA guides were selected from the Dolcetto CRISPRi library[71] with BsmBI compatible overhangs (Supplementary Data 3). These oligos were annealed, phosphorylated then ligated into BsmBI-digested pXPR050 (Addgene #9692), generating XPR-050-VHL. Monoclonal HEK293T LV-REPORT-12xHRE cell lines were transduced with the XPR-050-sgVHL virus, and stable cell lines selected with Puromycin. Subsequently, LV-SFFVp-CRISPRoffv2.1-IRES-BlastR or LV-EF1α-CRISPRoffv2.1-IRES-BlastR virus was infected into HEK293T LV-REPORT-12xHRE/XPR-050-sgVHL stable cells and selected with Blasticidin S (15 μg/ml) for 5 days. FACS was used to assess activation of the dFLASH-HRE reporter in parental (dFLASH-HRE/sgVHL) or CRISPRoffv2.1 expressing cells at day 5 or day 10 after Blasticidin S addition.

**Whole genome CRIPSR KO screens of the HIF pathway**
**Library preparation.** The Brunello whole genome KO library (Addgene #92379) was propagated in electrocompetent NEBstables by 5 parallel electroporations of 100 ng plasmid and 25 μl of electrocompetent NEBstables (lab made) using a Biorad Gene pulser (1.8 Kv, 200 mA, 25 μF) using 0.1 cm cuvettes (Biorad #165-2089), resulting in ~500x coverage. The Brunello plasmid library was purified using Machery Nagel endotoxin Free midi prep plasmid purification Kit (#740424.50).

Lentivirus was generated from 6 x T75 flasks as described above, collected in Optimem media (ThermoFisher; #31985088) and concentrated by ultracentrifugation at 50,000 g for 2 h at 4 °C using a Sorvall WX+ Ultracentrifuge Series centrifuge and swing out rotor with conical bottom tubes (#C14291 Beckman). Pelleted Lentivirus was resuspended in Optimem and frozen in aliquots at -80 °C. Lentivirus

was tired by seeding $1 \times 10^6$ cells into 6 well trays and infecting with serial dilutions of virus in the presence of 8 μg/ml polybrene. Two days later the media was replaced with 1 μg/ml Puro media and 5 days later the % survival of the cells was estimated.

**Inducible Cas9 expressing dFLASH cell lines**
dFLASH-HRE clonal lines in HEK293T and U2OS cells were used for whole genome CRISPR KO screens. First we generated inducible Cas9-TagBFP expressing lines by lentiviral infection of HEK293T-mcdFLASH-HRE or U2OS-mcdFLASH-HRE with LV-TET2BlastR-Cas9-TagBFP at a low MOI (<0.3) and selected in 10–15 μg/ml Blasticidin. HEK293T-mcdFLASH-HRE clonal LV-TET2Blast-Cas9-TagBFP cell lines were generated by treating cells with 400 ng/ml Doxycycline for 2 days prior to FACS sorting of single BFP positive cells into 96 well plates and subsequent screening of clones for Cas9-TagBFP expression using FACS. U2OS-mcdFLASH-HRE / LV-TET2Blast-Cas9-TagBFP cell lines were generated by two rounds of FACS enrichment of BFP high expressing cells following 400 ng/ml Dox treatment for 2 days.

**CRISPR KO validation by sgRNA nucleofection**
In order to validate inducible Cas9 mediated KO, we used a pooled sgRNA nucleofection protocol. Briefly, prior to nucleofection, cells were cultured in media containing 400 ng/ml dox for 2 days to induce Cas9-TagBFP expression. HEK293T cells were then resuspended in SF medium (Lonza, V4XC-2032) and U2OS cells in SE medium (Lonza, V4SC-1096). $2 \times 10^5$ resuspended cells were placed in each well of a 16-well nucleofection strip with 5 pmol sgRNA (Edit-R, HorizonDiscovery) for HEK293T cells and 6 pmol for U2OS cells and nucleofected using the Lonza 4D X nucleofection machine (AAF-1003X) with pulse code DS-150 used for HEK293T and CM-104 for U2OS cells. Cells were then plated into 2 wells of a 6 well plate containing media supplemented with 400 ng/ml dox. After 3 days, cells were treated with 0.5 mM DMOG / DMSO for 24 h (HEK293T) or 48 h (U2OS) and analysed by FACS. Control cells were suspended in SE/SF nucleofection media but did not contain a sgRNA and did not undergo nucleofection.

**Whole genome inducible KO screens**
Three replicate screens were performed in either HEK293T or U2OS mcdFLASH-HRE / LV-TET2Blast-Cas9-TagBFP cells using the Brunello whole genome KO library (Addgene #92379). For each replicate, 1.5-$2 \times 10^8$ cells in 10–15 T175 flasks were transduced with lentivirus at an MOI < 0.5 (500-1000x coverage), after 2 days the media was replaced with 1 μg/ml Puromycin and 10 μg/ml Blasticidin containing media, selected for ~1 week and maintained at ~500–1000x coverage. Cas9-TagBFP was induced with 400 ng/ml Dox for 5 days and cells were then treated with 0.5 mM DMOG for 1 (HEK293T) or 2 (U2OS) days. $6 \times 10^7$ - $1 \times 10^8$ cells were then reserved for library gDNA extraction and $1 \times 10^8$ FACS sorted for GFP +ve cells and either the lowest 5–8% or the highest 1–5% of Tomato cells. The library or FACS sorted cells were spun down at 400 g for 5 min at 4 °C and the pellet stored at -80°C prior to gDNA extraction.

**gDNA extraction and CRISPR library prep**
Cell pellets were resuspended in a minimum of 500 μl (per $5 \times 10^7$ cells) of digestion buffer (10 mM Tris-HCl pH 8.0, 150 mM NaCl, 10 mM EDTA, 0.1% SDS) supplemented with 200 μg/ml of proteinase K (Thermo EO0491) and 30 μg/ml of RNAase A (DNAase and RNAase Free, Thermo EN0531) and incubated at 37 °C for 3 h and then 55 °C overnight. Lysates were then mixed 1:1 with Phenol/chloroform/iso-amyl alcohol (PCI) solution (25:24:1) pH 7.8-8.2 (100 mM Tris) (Sigma 25666) and centrifuged at >16,000 g for 10 min at room temperature. The supernatant was transferred to a new tube and 0.3 vol of 24:1 chloroform/isoamyl alcohol solution (Sigma C0549) added to the phenol extracts and reextracted by vortexing and centrifuging. The

supernatants were combined and 1 µl of 15 mg/ml GlycoBlue (Thermo Scientific, AM9515) added with 0.5 vol of 7.5 M ammonium acetate (final concentration of 2.5 M). 0.7 Vol of Isopropanol was then added, mixed by inversion, incubated at room temperature for 5 min and centrifuged at 16,000 g for 30 min at 4 °C. Genomic DNA pellets were washed with 70% ethanol, centrifuged at >16,000 g for 5 min at 4 °C, air dried at 42°C, resuspended in 10 mM Tris pH 8.0 overnight at 42 °C and sheered by 5x -80 °C freeze thawing prior to PCR.

For PCR amplification, gDNA was divided into 50 µL reactions such that each well had at most 5 µg of gDNA. 0.2 µl of AmpliTaq Gold (1U, 4311820, Thermo) was added per 50 ul reaction with 2 mM MgCl₂ and 1 µl of 10 µM P5 F hU6 5′ TCTTGTGGAAAGGACGAAACACCG 3′ and P7 R sgRNA − 5′ TCTACTATTCTTTCCCCTGCACTGT 3′ primers in a PCR program of 95 °C for 3 min, and 30 cycles of 94 °C for 30 s, 58 °C for 30 s, 72 °C for 30 s, followed by extension at 72 °C for 10 min. The PCR reactions from each condition were combined and purified by SPRI bead (Beckman Coulter, B23317) purification, removing gDNA and primers. 10 ng of PCR product was then used in the next set of 10 x PCRs using 15 cycles and staggered barcoding primers to add sequencing adaptors. The PCRs were then pooled and purified by SPRI bead purification, quantified and sequenced on an Illumina Next-Seq500 1 x 75 bp high output run at the South Australia Genome Centre (SAGC).

### Pooled CRISPR screen analysis

Fastq files were trimmed and structure selected using fastq-grep https://github.com/dcjones/fastq-tools. Trimmed fastq files were then split by barcode using fastx_toolkit (version 0.0.14) fastx_barcode_splitter.pl function with the following parameters --mismatches 1 --bol --partial 1. Extracted sgRNA fastq files were then mapped to the Brunello reference library with Bowtie2 (version 2.5.1) and count files generated for subsequent screen analysis. To identify enriched or depleted sgRNA's and genes, count tables were analysed using MAGeck[72] and DEseq2 comparing high or low Tomato sorted pools vs libraries. Hits were defined as enriched statistically significant genes from MAGeck ($z > 0$ and $p < 0.05$) or DEseq2 lists ($LFC > 0.6$ and $p < 0.01$). We benchmarked dFLASH-HRE HIF CRISPR screens against 3xHRE- mCherry^ODD reporter CRISPR screen[9] by comparing the number of positively statistically ($p < 0.05$) enriched hits from dFLASH-HRE or published MAGeCK hit lists. Volcano Plots and bar plots comparing number of hits were generate in R using ggplot2.

### CRISPR knock-in of HA-Flag tags to endogenous HIF-1α and HIF-2α.

CRISPR targeting constructs targeted the region adjacent to the endogenous HIF-1α and HIF-2α stop codons[73]. Constructs were cloned into px330 (Addgene #42230) by ligating annealed and phosphorylated oligos with BbsI digested px330, using hHIF-1α and hHIF-2α CTD sgRNA (Supplementary Data 3). Knock-in of HA-3xFlag epitopes into the endogenous HIF-1α or HIF-2α loci in HEK293T cells was achieved by transfection with 0.625 µg of pNSEN, 0.625 µg of pEFIRES-puro6, 2.5 µg of px330-sgHIF-α CTD, and 1.25 µg of ssDNA HDR template oligo containing flanking homology to the CRISPR targeting site, the tag insertion and a PAM mutant into ~0.8 × 10⁶ cells using PEI (3 µg :1 µg). 48 h after transfection, the medium was removed from cells and replaced with fresh medium supplemented with 2 µg/ml puromycin for 48 h, before the cell medium was changed to fresh medium without puromycin. 48 h later cells were seeded by limiting dilution into 96-well plates at an average of 0.5 cells/well. Correct integration was identified by PCR screening using HIF-1α and HIF-2α gDNA screening primers (Supplementary Data 4). Positive colonies were reisolated as single colonies by limiting dilution. Isolated HIF-1α and HIF-2α tag insertions were confirmed by PCR, sanger sequencing and western blotting.

### High Content Imaging (HCI).

Cells were routinely seeded at $1 \times 10^4$ to $5 \times 10^4$ cells per well in black walled clear bottom 96 well plates (Costar

Cat#3603), unless otherwise stated. Cell populations were imaged in media at the designated time points at 10x magnification and $2 \times 2$ binning using the ArrayScan™ XTI High Content Reader (ThermoFisher). Tomato MFI and EGFP MFI was imaged with an excitation source of 560/25 nm and 485/20 nm respectively. Individual nuclei were defined by nuclear EGFP expression nuclear segmentation and confirmed to be single cells by isodata thresholding. Nuclei were excluded from analysis when they couldn't be accurately separated from neighbouring cells and background objects. Cells on image edges and abnormal nuclei were also excluded. EGFP and Tomato intensity was then measured for each individual nucleus from at least 2000 individual nuclei per well. Fixed exposure times were selected based on 10–35% peak target range. Image quantification was achieved using HCS Studio™ 3.0 Cell Analysis Software (ThermoFisher). For assessment of high throughput robustness of each individual reporting line in a high throughput setting (HTS-HCI), replicate 96 well plates were seeded for the HIF (10 plates), PGR (5 plates) and synFIH (3 plates) monoclonal reporter lines and imaged as above at 48 h. For the HIF line, each plate had 6 replicates per treatment (vehicle or DMOG) per plate. For the PGR line, 48 replicates per treatment, either vehicle or R5020 per plate. 24 replicates per treatment were also used for synFIH, with system robustness assessed between the DOX/DMSO and DOX/DMOG treatment groups. Z' and fold change (FC) for the Tomato/EGFP ratio for each individual plate was then calculated as per[35]:

$$Z' = 1 - \frac{(3\sigma_{c+} - 3\sigma_{c-})}{|\mu_{c+} - \mu_{c-}|}$$

Z' for every plate across each system was confirmed to be >0.5. Overall robustness of each system is the average of every individual Z' and FC for each independent plate, for each system. For temporal high content imaging, HIF, PGR and synFIH lines were seeded in plates and treated with requisite ligands immediately prior to HCI. Four treatments per plate were used to assess the synFIH monoclone (DOX, DMSO, DOX/DMSO, DOX/DMOG), with 100 ng/µL Doxycycline utilised, and 8 treatments per plate (vehicle, DMOG or R5020) were used to assess the PGR and HIF monoclonal lines. Plates were humidified and maintained at 37 °C, 5% CO₂ throughout the time course imaging experiment. Plates were imaged every 2 h for 40–48 h. At every timepoint, a minimum of 2000 nuclei were resampled from each well population.

### T47D mcdFLASH-PGR R5020 dose response curve and EC50 calculation.

T47D mcdFLASH-PGR cells were treated with increasing doses of 0.01-100 nM R5020 and quantified by HCI after 48 h. Tomato/GFP values were min/max normalised ($x' = \frac{(x'-x_{min})}{(x_{max}-x_{min})}$) within each experiment ($n = 3$) and the EC50 constant and curve fitted using the drc R package from ref. [74].

### Bimodal small molecule screen to identify activators or inhibitors of the hypoxic response pathway.

The library of natural and synthetic compounds was supplied by Prof. Ronald Quinn and Compounds Australia. 5 mM of each of the 1595 compounds were spotted in 1 µL DMSO into Costar 96 well (#3603) plates and stored at -80 °C prior to screening. Plates were warmed to 37 °C prior to cell addition. Monoclonal mcdFLASH-HIF HEK293T reporter cells were seeded at $0.5 \times 10^4$ cells per well across 20 Costar 96 black well clear bottom (#3603) plates pre-spiked with 5 mM of compound in 1 µL of DMSO in 100 µL. On each plate, 4 wells were treated with matched DMSO amounts to compound wells as were four 1 mM DMOG controls. Plates were then imaged using HCI (described above) at 36 h or 24 h for reporter activation. Wells were then treated with 100 µL of 2 mM DMOG (for 1 mM DMOG final, 200 µL media final). Four vehicle and 8 DMOG-treated controls (excluding the initial controls from the activator screen) were used for the inhibitor screen. Cells were imaged again in 36 h (Screen 1)

or 24 h (Screen 2) after treatment with 1 mM DMOG in the compound wells. Data was Z scored and compounds within ±2 SD (-2 < Z < 2) of the EGFP MFI were classified as having unchanged transcriptional effects. Compounds with a Tomato/EGFP ratio >2 SD and unchanged GFP were included as a putative activator hit. For the inhibitor screen, compounds with a Tomato/EGFP ratio lower than -2SD (Z < 2) from the normalised population were considered putative inhibitors. Pearson correlations were then used to compare Tomato/GFP expression between screens and compound treatments with the base R package (4.4.0). Putative activators and inhibitors identified in the screens were re-spotted at 1 mM, 2.5 mM and 5 mM in 1 µL of DMSO in Costar Cat#3603 96 well trays. Activators were rescreened by HCI after 24 h against $1 \times 10^4$ HEK293T mcdFLASH-HIF cells in biological duplicate with vehicle and 1 mM DMOG controls in 100 µL. Inhibitors were rescreened by HCI after 24 h in duplicate against $1 \times 10^4$ HEK293T mcdFLASH-HIF cells with 1 mM DMOG added to compound wells. Final compound concentrations were 10 µM, 25 µM and 50 µM respectively with Tomato MFI and Tomato/EGFP ratio for each compound assessed. dFLASH Bimodal high throughput screen details can be found in Supplementary Table 2.

**Flow cytometry analysis and sorting (FACS).** Prior to flow cytometry, cells were trypsinised, washed in PBS and resuspended in flow cytometry sort buffer ($Ca^{2+}/Mg^{2+}$-free PBS, 2%FBS, 25 mM HEPES pH 7.0 or PBS with 2% FBS), followed by filtration through a 40 µM nylon cell strainer (Corning Cat#352340). Cell populations were kept on ice prior to sorting. Flow cytometry was performed either using the BD Biosciences BD LSR Fortessa or a BD Biosciences FACS ARIA2 sorter within a biosafety cabinet under aseptic conditions, using an 85 µM nozzle. Cell populations were gated by FSC-W/FSC-H, then SSC-W/SSC-H, followed by SSC-A/FSC-A. EGFP fluorescence was measured by a 530/30 nm detector, and the Tomato fluorescence was determined with the 582/15 nm detector. A minimum of 10,000 cells were analysed per sample. Data is presented as $\log_{10}$ intensity for both fluorescent proteins. Tomato induction was gated from the top 1% of the negative control population. Cell counts for histograms are normalised to mode unless stated otherwise. FACS analysis was done on FlowJo™ v10.9.1 software (BD Life Sciences). Representative examples of the gating strategies used is provided in supplementary Fig. 11.

**Primary mouse granulosa cell culture and dFLASH transduction.** CBAF1 female mice were stimulated with 5 IU of pregnant mare serum gonadotrophin (PMSG) (Medix Biochemica, 493-10-10) and culled 44 h post-PMSG. Ovaries were then dissected, and granulosa cells isolated with all cumulus-oocyte complexes removed. Granulosa cells were counted, then seeded at 250,000 cells/well in a 6 well plate. Cells were cultured in a 1 to 1 mix of DMEM (no glucose; Gibco, Thermo scientific 11966025) to F-12 nutrient mix (Gibco, Thermo scientific 31765035) with 5% FBS, 35 nM testosterone (Sigma Aldrich D4902) and 1 µM retinoic acid (Sigma Aldrich R2625) at 37 °C, 5% $CO_2$. 2 h post seeding, media was removed from all wells to remove debris and replaced with 2 ml of fresh culture media. The next day, cells were transduced with $5 \times 10^6$ IU dFLASH-HRE virus per well. Four days later, media on cells was replaced with fresh culture media containing 50 µg/mL hygromycin B (ThermoFisher #10687010), with cells kept under selection for 2 days. All data is representative of 3 biological replicates, where each replicate consists of granulosa cells pooled from 2 independent mice.

**dFLASH-HRE mouse granulosa cell high content imaging and FACS analysis**
80–90% confluent dFLASH-HRE stable mouse GCs were washed with PBS prior to adding 500 µL of TrypLE (Gibco, ThermoFisher 12604013) and placing at 37 °C for 10–15 min to detach and dissociate. Cells were resuspended in culture media, spun down and resuspended in fresh media, then counted and seeded at 20,000 cells/well into black-

walled, clear bottom 96 well plates (Costar 3603) for high content imaging or 6 well plates at 35,000 to 95,000 cells/well for flow cytometry analysis.

High content imaging: The next day, cells were gently washed to remove debris before being treated with 50 µM FG-4592 or vehicle (0.1% DMSO). 24 h post treatment, media in each well was removed and replaced with 100 µL of FluroBright DMEM (Gibco, Thermo Scientific A1896701) for imaging. Cells were imaged using the Thermo Fisher ArrayScan XTI High Content Reader as described for all immortalised cell lines, but with an EGFP excitation of 0.8 sec, tomato excitation of 0.5 sec and counting 1000 cells/well.

Flow cytometry: Post seeding, cells were left to grow in 6 well plates for 1 week or until ~60% confluent, then treated with 50 µM FG-4592 or vehicle (DMSO). 24 h post treatment, cells were suspended using TrypLE in culture media, pelleted, then resuspended in PBS + 2% FBS for flow cytometry analysis using a BD Biosciences LSRFortessa. Cells were gated by FSC-A/SSC-A, then FSC-H/SSC-W, followed by SSC-H/FSC-W to select for single cells. Tomato mean fluorescence intensity was then quantified in GFP positive cells and graphed for each independent replicate. 10,000 cells total were analysed per sample using the detectors described in the Flow Cytometry methods section. Data analysis was completed using FlowJo™ v10.9.1 software (BD Life Sciences).

**Reverse transcription and real time PCR.** Cells were seeded in 60 mm dishes at $8 \times 10^4$ cells per vessel and left overnight before treatment for 48 h with 1 mM DMOG or 0.1% DMSO. Cells were lysed in Trizol (Invitrogen, Thermo Scientific 15596026), and RNA was purified with Qiagen RNAEasy Kit (#74104), DNaseI treated and reverse transcribed using M-MLV reverse transcriptase (Promega M1701). cDNA was then diluted for real time PCR. Real-time PCR used primers specific for *HIF1A*, and human RNA Polymerase 2 (*POLR2A*) (Supplementary Data 4). All reactions were done on a StepOne Plus Real-time PCR machine utilising SYBR Green, and data analysed by 'QGene' software. Results are normalised to *POLR2A* expression. RT-qPCR was performed in triplicate and single amplicons were confirmed via melt curves.

**Time lapse spinning disc confocal microscopy.** HEK293T mcdFLASH-HIF and T47D mcdFLASH-PGR cells were seeded at $1 \times 10^5$ or $5 \times 10^5$ cells per dish respectively, into 50 µg/mL poly-D-lysine coated 35 mm high glass bottom dishes (Ibidi, #81158) in FluoroBrite DMEM (Gibco, A1896701) with 10% FBS, 1% pen/strep, 1% Glutamax and 10 mM HEPES pH 7.9, and incubated overnight at 37 °C with 5% $CO_2$ prior to imaging. Cells were treated with either 0.5 mM DMOG (mcdFLASH-HIF) or 100 nM R05020 (mcdFLASH-PGR) immediately prior to imaging with a CV100 cell voyager spinning disk confocal microscope for Tomato (561 nm, 50% laser, 400 ms exposure and 20% gain) and EGFP (488 nm, 50% laser, 400 ms exposure and 20% gain) fluorescence for 48 h post treatment in 15 min imaging intervals. Images were captured with a 40x objective lens with a ~30 µm Z stack across multiple fields of view. Maximum projected intensity images were exported to Image J for analysis and movie creation.

**Cell lysis and immunoblotting.** Cells were washed in ice-cold PBS and lysates were generated by resuspending cells in either cell lysis buffer (20 mM HEPES pH 8.0, 420 mM $NaCl_2$, 0.5% NP-40, 25% Glycerol, 0.2 mM EDTA, 1.5 mM $MgCl_2$, 1 mM DTT, 1x Protease Inhibitors (Sigma P8340)) (Supplementary Fig. 5) or urea lysis buffer (6.7 M Urea, 10 mM Tris-Cl pH 6.8, 10% glycerol, 1% SDS, 1 mM DTT) (Fig. 7, Supplementary Fig. 9). Protein quantification was achieved by Bradford Assay (Bio-Rad 5000006). Lysates were separated on a 7.5% SDS-PAGE gel, transferred to nitrocellulose via TurboBlot (Bio-Rad) and probed with the following primary antibodies; anti-HIF1α (1:1000, BD Biosciences #810959), anti-HA (1:1000, HA.11, Biolegend #16B12), anti-Tubulin (1:10,000, Serotec #MCA78G), anti-GAPDH (1:5000, Sigma #G8796)

and anti-ARNT (1:2000, Proteintech #14105-1-AP). Primary antibodies were detected using horseradish peroxidase conjugated secondary antibodies (Pierce Bioscience, anti-mouse #31430, anti-rat #P0450). Blots were visualised via chemiluminescence and developed with Clarity Western ECL Blotting substrates (Bio-Rad 1705061).

**In vitro iron chelation activity assay.** Chelation of iron for RQ200674 was measured by a protocol adapted from[75] for use in a 96 well plate format. 0.1 mM $FeSO_4$ (50 µL) and 50 µL of RQ200674, Dipyridyl (Thermo Scientific, 117500100) (positive control) or DMOG solutions were incubated for 1 h at room temperature prior to addition of 100 µL of 0.25 mM Ferrozine (Sigma 160601) and incubated for a further 10 min. Absorbance was measured at 562 nM. Chelation activity was quantified as:

$$Chelation\ activity = \frac{(A_{control} - A_x)}{A_{control}} \times 100$$

Where $A_{control}$ is absorbance of control reactions without RQ200674, DP or DMOG and $A_x$ is absorbance of solutions with compound.

**Statistics and reproducibility.** All data in graphs are presented as a mean ± sem unless otherwise specified. Significance was calculated by a Two-Way ANOVA with Tukey's multiple comparison or unpaired $t$-test with Welches correction where appropriate using Graphpad PRISM (version 9.0.0). No statistical method was used to predetermine sample size, no data were excluded from the analyses, the experiments were not randomized and the investigators were not blinded to allocation during experiments and outcome assessment. All statistical analysis is from three independent experiments.

**Figure creation.** Schematics and diagrams were created with BioRender (BioRender.com) and graphs were made either with ggplot package in R[76] or GraphPad PRISM (version 9.0.0).

**Reporting summary**
Further information on research design is available in the Nature Portfolio Reporting Summary linked to this article.

## Data availability
All data generated in this study are provided in the Source Data File. Plasmid sequences are available from Addgene (https://www.addgene.org/David_Bersten/). The raw CRISPR sequencing data generated in this study have been deposited in the GEO database under accession code GSE290538. Source data are provided with this paper.

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

## Acknowledgements

We thank Nicholas Smith, Alexander Pace, and members of our laboratories for critical feedback and helpful discussions. We also wish

to acknowledge Adelaide Microscopy and the AHMS and SAHMRI Flow Cytometry facilities for technical assistance. We acknowledge Compounds Australia (www.compoundsaustralia.com) for their provision of specialized compound management and logistics research services to the project. This work was supported by Australian Government Research Training Scholarships (T.P.A., A.E.R.), The Emeritus Professor George Rodgers AO Supplementary Scholarship (T.P.A., A.E.R.), The Playford Memorial Trust Thyne Reid Foundation Scholarship (A.E.R), The George Fraser Supplementary Scholarship (A.E.R.), The University of Adelaide Biochemistry Trust Fund (D.J.P. and M.L.W) and the Bill and Melinda Gates Foundation Contraceptive Discovery Program [OPP1771844] (D.C.B, D.L.R).

## Author contributions

Study was initially conceived by D.C.B. and M.L.W. T.P.A., D.C.B., A.E.R. designed and performed experiments. T.P.A, D.C.B., M.L.W., M.L and R.J.Q. performed and analysed the bimodal drug screening campaign. M.R. and A.E.R. derived the FIH KO cell line. D.C.B. and A.E.R. conceived, performed and analysed the whole genome CRISPR screens. Primary cell experiments were conceived, designed and performed by A.E.R, D.L.R., and A.E.S. T.P.A., D.C.B., A.E.R., D.J.P., D.L.R. and M.L.W. wrote the manuscript with input from all authors. Work was supervised by D.C.B., D.J.P., D.L.R. and M.L.W.

## Competing interests

The authors declare no competing interests.
