## [Transparent Peer Review file · Nature Communications]

dFLASH; dual FLuorescent transcription factor Activity Sensor for Histone integrated live-cell reporting and high-content screening

Corresponding Author: Dr David Bersten

Version 0:

Reviewer comments:

Reviewer #1

(Remarks to the Author)

Allen et al present work developing fluorescent reporters as an optimised system for drug screening for activators or inhibitors. It is essentially a methods paper and the methodology is clearly described. The benefit of a modular system and having an internal fluorescent control is emphasised. It does seem a useful system but it is less clear whether dFLASH offers a substantial advantage compared to other screening systems (e.g. with luciferase) that are typically done at higher throughput for compound screens. TF fluorescent reporters have been used in various forms for a long time. The screens presented show the potential of the approach but the compound library has not yet yielded drugs that would really be classed as TF inhibitors, which is a current challenge for HIF1.

Specific points:

- The actual number of responsive elements used in the main reporter systems is not clearly explained. 12X HRE seems excessive and implies that the system may not be as sensitive as claimed. The concern here is that a high number of REs becomes less biologically relevant, and this may be important when considering drug screens. The other reporters seem to only have 5X RE copies. The correct diagrammatical representation of the REs should be included in Figure 2.
- The time taken to activate the HIF reporter seems slow, at around 10 hrs. HIF should be stabilised within minutes and HIF targets should increase at the message level within a few hours. Why is this induction slower? This may be a concern for drug screening.
- None of the dFLASH-HRE experiments seem to have been done in different oxygen gradients. Are similar effects seen? The use of DMOG will inhibit both PHDs and FIH but do specific PHD inhibitors activate the reporter similarly?
- Regarding the drug screens, while there is good concordance with compounds that activate, there is more variability in the inhibitor screens between 24 and 36 hours. As there are already clinically useful drugs to activate HIFs, the challenge (as stated in the manuscript) is to find specific inhibitors. This variability may relate to the stability of Tomato and GFP, which is different from HIF. Therefore, how dynamic is this system? E.g. how long does it take for Tomato levels to return to baseline once DMOG is washed out?
- The dFLASH-PRE seems a more sensitive reporter system than dFLASH-HRE. Accept it is likely beyond the scope of this work but it would be helpful to have some indication from the authors as to the utility of this reporter in chemical screens.
- How was the drug toxicity assessed in the small molecule screens? Were the cells stained for cell death/apoptosis, or was it judged by GFP signal/cell morphology? GFP alone may not be a good indication of toxicity as it is likely to be very stable.

Reviewer #2

(Remarks to the Author)

The work by Allen et al. presents dual Fluorescent transcription factor Activity Sensor for Histone integrated live-cell reporting (dFLASH), a modular sensor for TF activity that can be readily integrated into cellular genomes. The authors demonstrate that dFLASH platforms can sense the regulation of endogenous Hypoxia Inducible Factor (HIF) and Progesterone receptor (PGR) activities, as well as regulated coactivator recruitment to a synthetic DNA-Binding Domain-Activator Domain fusion proteins. Finally, the authors test the utility of this platform for functional genomics applications by using CRISPRoff to modulate the HIF regulatory pathway, and for drug screening by using high content imaging in a bimodal design to isolate activators and inhibitors of the HIF pathway.

The manuscript is well written, the experiments are very thoroughly executed and the proposed tool is certainly of great interest for the field. However, the impact of the study is likely to increase by addressing the following questions experimentally:

Major:

Did the authors address the degree of cell death and apoptosis after integration of dFLASH constructs? In this regard, it would also be important to test the dFLASH in non-transformed, non-immortalized and non-cancerous cells. Would the dFLASH tool work in primary cells?

Especially with regard to the studies on the HIF pathway, it would be important to correlate the reporter kinetics after DMOG treatment shown in figure 2 with HIF protein levels and eventually mRNA expression of (a) HIF target gene(s). Along these lines and despite the potential issues with fluorescent reporters in low oxygen conditions, how would the dFLASH reporter signal look like in a more physiological hypoxia/reoxygenation experiment?

Minor:

The figure organization could benefit from moving some of the data from the suppl. figures to the main figures. Lots of data that are required to understand the manuscript are shown in the suppl. figures while the main figures are rather sparse in panels.

Reviewer #3

(Remarks to the Author)

dFLASH; Dual Fluorescent transcription factor Activity Sensor for Histone integrated live-cell reporting and high-content screening

Comments to the authors

Allen et al report the development of dFLASH platform that is a modular transcription factor sensor that can be used in live cell reporting for multiple downstream applications such as functional genomics screens, drug discovery and synthetic biology. This system consists of two dual color reporters (an internal control nucEGFP as well as dTomato reporter as a readout). The authors demonstrate the use of this reporter system for readout of endogenous HIF and PGR related pathways. Together these authors developed a system that can be used for a variety of applications. Major strengths of this system are that it can be applied to multiple cell types and the modular design also allows researchers to study various endogenous processes. The dual reporter system also allows readouts through flow cytometry or imaging. Furthermore, they successfully used this system to identify novel inhibitors and activators of the HIF pathway in screens. However, it would be useful if the authors could perhaps address more strongly, the ways in which their system is advantageous or different from previous reporter systems. Alternatively, the authors could provide a comparison of previous reporter systems to better communicate the impact or uniqueness of their system. Second, adding more quantifications to some of the imaging and flow cytometry data could also strengthen the results. Overall the data presented in this manuscript is convincing and the applications of the reporter system are strongly demonstrated. Therefore, it could provide an amenable system for reporter-based screening applications to study other endogenous pathways as well.

Additional comments:

1. Within the introductions and discussions, the authors could discuss previous systems that were built for TF sensing, for example luciferase or reporter systems have been used in the past to enable HIF1a pathway analysis (See a). Discussion of these systems alongside the one that the authors have built would be useful in communicating the impact of the developed reporter system. Additionally, the authors could also determine how their system addresses some of the shortcomings of the previous systems and the types of pathways that can be analyzed by the system in order to make the discussion more comprehensive.

a. An optimized reporter of the transcription factor hypoxia-factor inducible factor 1a reveals complex HIF-1a activation dynamics in single cells [https://www.jbc.org/article/S0021-9258\(23\)00241-7/fulltext](https://www.jbc.org/article/S0021-9258(23)00241-7/fulltext)

2. In Figure 1B, the authors are showing how their built reporter system can be used for imaging-based analysis. This figure can be strengthened if they include the pixel intensities of the GFP and RFP channels from select regions of the microscopy images. More specifically it would be helpful to see that the RFP increases and the GFP remains constant. It would also help account for the background fluorescence of the Tomato reporter in using the imaging application. Finally, they could also consider adding black/white image panels.

3. The schematic in Figure 1B is a little misleading, can the authors include more details or enhance the figure to guide the

reader? For example, add +DMOG etc. or potentially remove the arrow.

4. Within the flow plots shown in Figure 2, D-F it would help if the authors quantified the percentage of cells that respond to the given treatments. For example, they can show their gating strategies. They should also comment on the fact that these are not clonal lines and are a heterogeneous population, since they are following up on these experiments with single cells clones.

5. Could the authors explain why some of their cells do not express dTomato despite antibiotic selection? A two-axis flow plot with gating may be more comprehensive for the main figure (like those shown in Supp Fig2). For example, their flow plots can indicate a range of fluorescence intensities in the tomato channel, how would these numbers change if the authors gated for the highest GFP expressing cells (where the reporter is integrated in euchromatin region). Would that in turn affect the Tomato readout?

6. Figure 4C/Supplementary Fig 6: The authors can include a single cell flow plot and gating information in the figure to make it more comprehensive for the reader (including numerical/gating details e.g. % of cells).

7. The authors could clarify the reasoning behind using different promoter versions for CRISPRoffv2.1

8. It appears that the clonal cell lines in Figure 4 produce multiple dTomato peaks, does that indicate that the line is slightly leaky for expression of dTomato? How does it compare to no fluorescence - doing an overlay would be helpful? Could potentially be a challenge for use in screens if there is indeed background.

9. In the CRISPRoff experiment, did the authors also include a non-targeting guide? Might be useful to show or include as control

10. Not essential but the authors can confirm VHL protein knockdown in cells post CRISPRoff induced silencing using a western blot (shows sensitivity of the system).

11. Discuss how this reporter system can be of better use compared to current systems for screening methods

12. For Figure 5, the authors could describe the Activation Screen and Inhibitor with more detail or clarity within the text such that it is easy to follow, or describe the workflow and expected results with more clarity.

13. For Figure 5B/C, the authors could label the top hits from their screen in the plot

14. For Figure 5B, the squares, circles and triangles can be a little difficult to distinguish, the authors could consider using different colors for clarity.

Version 1:

Reviewer comments:

Reviewer #1

(Remarks to the Author)

Overall, the authors have not adequately addressed the issues raised during the initial submission, and the manuscript has not been significantly improved. There does not seem to be a much of an advance here.

- Novelty of the HRE reporter. The authors present a table comparing some prior HIF reporters, which is selective and not accurate. Other HRE-fluorescent reporters exist and have been described over the past 20 years (e.g. PMID 11774035, 17270179, 26598532). Some have dual colors, destabilised fluorescent proteins (PEST or adding the ODDD), and have been shown to work in cell lines and in vivo settings. The HRE-HIF-ODD reporter is not constitutively active (minimal SV40 promoter x3HRE). The addition of the internal fluorescent control remains the only novelty of the reporter, but the published NanoFIRE manuscript now referred to in the authors' response (describing a similar approach to dFLASH but with luciferase) already details the use of the 12XHRE dFLASH fluorescent system.

- The authors state that endogenous enhancers contain 5-6 sites for TFs, but still use 12 HREs. This necessity for 12x HRE is at odds with other prior reporters (e.g. HypoxCR).

- The variability in the inhibitory screens remains a concern. The stability of Tomato is not an advantage as stochastic/basal induction of the reporter will occur (as noted), and dFLASH will not detect turning a TF off. Same argument applies for reoxygenation.

- The assays of cell death or drug toxicity have not been adequately addressed.

- The HRE screen has so far not identified any interesting results.

- The utility of the system in other cell types (SFig. 3D) is not that impressive, with a fairly minimal induction.

Reviewer #2

(Remarks to the Author)

The authors have addressed all concerns in the revised manuscript. Congratulations!

Reviewer #3

(Remarks to the Author)

The authors have successfully addressed all the questions and made modifications to the manuscript accordingly.

Version 3:

Reviewer comments:

Reviewer #4

(Remarks to the Author)

I was asked primarily to adjudicate the different views of three reviews of this paper and its revision. In my view, the authors have done an excellent job in addressing the comments, and I feel that the comments of Rev 1 are asking too much. Yes, reporters have been developed before including around hypoxia, but the internal control **does** add a lot of value, as does nuclear localization, and the framework appears to be more broadly useful for reporters in general, which to me seem to often be put out into the world in haphazard and poorly controlled ways. I also don't think it's fair to knock the authors for citing their own preprint. The Table that the authors provide now (which I assume will be included in the paper either as main or supplement) is useful and transparent, and to my knowledge accurate (and I say this as the developer of one of the methods listed there, which absolutely has limitations that are fairly noted there). Both imaging and non-imaging based reporter methods need better controls, and the internal control is a terrific way to go. I might suggest that the authors also add scQers (Nature Methods 2024) to the table, which has a sequence-based internal control that I think reinforces the broader point about how important this is in any context where one is looking at single cells rather than bulk (in this case by imaging, in that case by sequencing).

dFLASH response to reviewers

REVIEWER COMMENTS

Reviewer #1 (Remarks to the Author):

Allen et al present work developing fluorescent reporters as an optimised system for drug screening for activators or inhibitors. It is essentially a methods paper and the methodology is clearly described. The benefit of a modular system and having an internal fluorescent control is emphasised. It does seem a useful system but it is less clear whether dFLASH offers a substantial advantage compared to other screening systems (e.g. with luciferase) that are typically done at higher throughput for compound screens. TF fluorescent reporters have been used in various forms for a long time.

We thank reviewer #1 and #3 for the request for a clearer explanation of the advantages of the dFLASH system over other analogous approaches.

After discussion with the editor and comparison with analogous fluorescent reporter systems we concluded that no other systems have been developed that can be directly benchmarked in the experimental paradigms outlined here. We have specifically excluded luciferase systems from this as they cannot be applied to live cell high content or FACS based analysis/enrichment as described here. We have developed and published a companion bioluminescent system called NanoFIRE (doi: 10.3390/biom13101545.), that like the dFLASH system robustly reports (20-150x induction, $Z' > 0.5$) on endogenous and synthetic TF pathways in stable cell lines.

The automated nuclear segmentation enabled by the dual nuclear fluorescent proteins as outline by the dFLASH system allows high-throughput normalised high-content imaging not enabled by other systems.

Moreover, we optimised and benchmark the dFLASH system in a variety of ways in the paper including downstream promoter composition and the length of the linker between the enhancer and promoter all which allows a robust reporting platform. Further, we benchmark the utility of the system against two endogenous unrelated TF pathways and one synthetic reporter system using robust screening metrics (Z') which evaluate the screen performance. This is not often observed even in large scale drug screening papers but commonplace in more industry-focused settings.

We agree that specific discussion of the advantages, optimisation and benchmarking of the dFLASH system has not been clearly outlined in the paper and as such we have made some changes to the text. (Page #12 Paragraph#1 line #722-782).

The screens presented show the potential of the approach but the compound library has not yet yielded drugs that would really be classed as TF inhibitors, which is a current challenge for HIF1.

We agree with the sentiment of Reviewer #1 who outlines that true HIF1 inhibitors are a significant challenge and highly sort after. We describe a proof of principle small scale screen to demonstrate the utility of the system to both identify positive

and negative regulators in a single experimental paradigm that could act directly or indirectly on the HIF pathway. Extending from this work, the dFLASH system is currently being applied to much larger (>300,000 compound screen) by independent drug screen laboratories reporting similar robust screening metrics presented here. Unfortunately, this is beyond the scope of this paper.

Specific points:

- The actual number of responsive elements used in the main reporter systems is not clearly explained. 12X HRE seems excessive and implies that the system may not be as sensitive as claimed. The concern here is that a high number of REs becomes less biologically relevant, and this may be important when considering drug screens. The other reporters seem to only have 5X RE copies. The correct diagrammatical representation of the REs should be included in Figure 2.

This has now been corrected in the schematic for Figure 2. Number of repeats and schematic of a 12xHRE and the 5xPRE & 5xGRE are now displayed for reader clarity, attached below.

Below we address the reviewers comments on the sensitivity of the dFLASH reporter system:

We set out to design a sensitive dual fluorescent reporter that allows use in high-throughput screening via FACS or by high content image segmentation and quantification. We aimed to generate systems which retained endogenous signalling mechanisms but provided robust and consistent signal-to-noise upon activation, as this is a fundamental requirement of screen performance. This remains a challenge however, and is currently being addressed by alternative approaches (see Lampson et al Cell 2024; <https://doi.org/10.1016/j.cell.2024.03.022>). To address this, our 12xHRE response element consists of HIF enhancers from LDHA, ENO1 and PGK1, which are well characterised endogenous HIF target genes, ensuring the endogenous signalling mechanisms of HIF are maintained. By repeating response element 12 times, this provided a reporter system with robust signal to noise which is required for screening applications, but maintained the endogenous signalling mechanism and sensitivity. We have changed the text and methods section to more clearly state this. We found in our Gal4 and PRE controlled systems that 5x response elements (also obtained from endogenous target gene enhancers for the PRE) were sufficient for superior dFLASH reporter induction, indicating that a large number of response elements is not a requirement for sensitive reporter induction in these contexts.

However, a number of historical (<https://doi.org/10.1002/j.1460-2075.1992.tb05607.x> and [https://doi.org/10.1016/0022-2836\(90\)90187-Q](https://doi.org/10.1016/0022-2836(90)90187-Q)) and recent (<https://doi.org/10.1016/j.cell.2012.12.027> and <https://doi.org/10.1101/2024.02.02.578660>) papers demonstrate that gene activation does not scale linearly with the number of enhancer elements increasing sharply and plateauing with up to ~7-8 repeats. This indicates that while synergism is necessary for robust reporting this is unlikely to be increase beyond 7-8 repeats. We also note that on average endogenous enhancers contain 5-6 binding site for TFs, (<https://doi.org/10.1038/s41586-020-2528-x>) indicating that designs implemented here are inline with those of native enhancers.

The time taken to active the HIF reporter seems slow, at around 10 hrs. HIF should be stabilised within minutes and HIF targets should increase at the message level within a few hours. Why is this induction slower? This may be a concern for drug screening.

The reviewer is correct to outline that the reporter induction begins at ~10hrs post activation. However, while HIF is stabilised within minutes to hours (<https://doi.org/10.1016/j.jbc.2023.104599> and <http://doi.org/10.1126/science.1059796>), reporter output is a more complex function of the time taken for accumulation of nuclear HIF, DNA binding and transcriptional activation dynamics as well as the Tomato protein translation and folding dynamics. Indeed, peak HIF protein is often observed 4-6hrs post induction and this has been shown to be cell type dependent with altered induction dynamics.

As many HIF target gene proteins do not appear until ~8-16hrs post induction¹ we view a 10hr lag prior to population Tomato fluorescence is in line with native HIF

target gene dynamics. However, it is noted that as seen in Supp Video 2 the induction of the monoclonal reporter lines in single cells is heterogenous and likely reflects stochastic mechanisms of gene activation, this is averaged in population dynamics in Figure 3.C.

None of the dFLASH-HRE experiments seem to have been done in different oxygen gradients. Are similar effects seen? The use of DMOG will inhibit both PHDs and FIH but do specific PHD inhibitors activate the reporter similarly?

We have now added new data with the experiments suggested by Reviewer #1. We show similar levels of induction under hypoxic conditions (~1%) (Figure 3f) and using the Prolyl hydroxylase (PHD) specific inhibitor FG-4592 (Supplementary Figure 6c) we see a difference in activity between DMOG (a pan-2-oxoglutarate dependent dioxygenase inhibitor) and PHD-specific FG-4592. Both upregulate the reporter, however FG-4592 does so less than DMOG. This has been added to the result text on Page 9, Paragraph 2, Line 584-587.

-Regarding the drug screens, while there is good concordance with compounds that activate, there is more variability in the inhibitor screens between 24 and 36 hours. As there are already clinically useful drugs to activate HIFs, the challenge (as stated in the manuscript) is to find specific inhibitors. This variability may relate to the stability of Tomato and GFP, which is different from HIF. Therefore, how dynamic is this system? E.g. how long does it take for Tomato levels to return to baseline once DMOG is washed out?

The bimodal screening methodology was primarily to outline the utility of live-cell drug screens in assaying multiple time points. We acknowledge that the consistency may be improved by single time-point screens. In saying this, the in-screen Z' remained >0.5 (mean of 0.62 for the DMOG 36-hour screen and 0.61 for the DMOG 24-hour screen across all 20 plates), indicating that the assay was highly consistent at the later time points. Additionally, independent plate controls have now been added to Supp Figure 8.

The dFLASH system was designed such that Tomato is not regulated by O₂ dependent proteasomal degradation as the native HIF system is, allowing relatively consistent quantitation of the transcriptional output. The however GFP contains a PEST element such that it has a protein turnover of ~ 2hrs (also discussed below).

-The dFLASH-PRE seems a more sensitive reporter system than dFLASH-HRE. Accept it is likely beyond the scope of this work but it would be helpful to have some indication from the authors as to the utility of this reporter in chemical screens.

Yes, this is indeed an exciting prospect for identification of non-hormonal drugs that might target PGR. To demonstrate the utility of this reporter we have since performed whole genome CRISPR screens identifying (>261 genes with a p-value < 0.001) new regulators of the Progesterone response pathway. We believe this outlines the high-quality of the dFLASH system in a variety of screening approaches,

but is indeed beyond the scope of this paper. We provide some data from these screens to validate their use to the reviewer but not within the paper. Indeed, we identify PGR as the top hit from this dFLASH loss of function CRISPR screen as well as other known (NCoA2) and many novel regulators.

[REDACTED]

- How was the drug toxicity assessed in the small molecule screens? Were the cells stained for cell death/apoptosis, or was it judged by GFP signal/cell morphology? GFP alone may not be a good indication of toxicity as it is likely to be very stable.

The EGFP expressed within dFLASH is destabilised (contains a PEST element) and therefore protein turn-over occurs every ~2 hours. Due to this rapid turn-over, we believe the PEST-EGFP used within dFLASH does provide a better readout of toxicity (i.e. identification of a decrease in expression), compared to if standard EGFP is used, and therefore we use this change in EGFP as a read out of toxicity in the dFLASH system. However, we do agree with the reviewer that thorough assessment of toxicity in high throughput screens is an important consideration, and we are currently considering alternative approaches for future works.

Reviewer #2 (Remarks to the Author):

The work by Allen et al. presents dual FLuorescent transcription factor Activity Sensor for Histone integrated live-cell reporting (dFLASH), a modular sensor for TF activity that can be readily integrated into cellular genomes. The authors demonstrate that dFLASH platforms can sense the regulation of endogenous Hypoxia Inducible Factor (HIF) and Progesterone receptor (PGR) activities, as well as regulated coactivator recruitment to a synthetic DNA-Binding Domain-Activator Domain fusion proteins. Finally, the authors test the utility of this platform for functional genomics applications by using CRISPRoff to modulate the HIF regulatory pathway, and for drug screening by using high content imaging in a bimodal design to isolate activators and inhibitors of the HIF pathway.

The manuscript is well written, the experiments are very thoroughly executed and the proposed tool is certainly of great interest for the field. However, the impact of the study is likely to increase by addressing the following questions experimentally:

Major:

Did the authors address the degree of cell death and apoptosis after integration of dFLASH constructs? In this regard, it would also be important to test the dFLASH in non-transformed, non-immortalized and non-cancerous cells. Would the dFLASH tool work in primary cells?

Cell lines were integrated with dFLASH reporters by lentivirus transduction at a low MOI (<0.3) to ensure that each cell contained ~1 integrant per cell as described in the methods. After hygromycin selection all most all of the GFP +ve cells survived. We did not directly assess the degree of cell death and apoptosis after integration; as expected cell death was observed after transduction and subsequent selection due to death of non-transduced cells. However, once a stable line was established, minimal subsequent cell death is observed and cells tolerate the dFLASH reporter well for long periods of culture.

We have subsequently tested a number of non-transformed or primary cells lines with the dFLASH-HRE system and demonstrate significant FG-4592 dependent induction of the reporter in primary mouse granulosa cells. This has been included in Supp Figure3 and discussed on page # 7 paragraph 2, lines 362-370. However, we acknowledge that substantial challenges remain for the application of the dFLASH system in primary / non-immortalised cells. In some contexts that we tested, such as human embryonic stem cells, one, or a combination of the constitutive control EGFP promoter, the destabilised EGFP and viral construct silencing made it difficult to implement the dFLASH reporter. Thus, it is likely that implementation of dFLASH across primary and non-immortalised cells will to require cell-type specific optimisation (such as changing the control promoter or using non-destabilised EGFP), which we are currently attempting in future efforts.

Especially with regard to the studies on the HIF pathway, it would be important to correlate the reporter kinetics after DMOG treatment shown in figure 2 with HIF protein levels and eventually mRNA expression of (a) HIF target gene(s). Along these lines and despite the potential issues with fluorescent reporters in low

oxygen conditions, how would the dFLASH reporter signal look like in a more physiological hypoxia/reoxygenation experiments

For discussion of dFLASH response to changes in oxygen levels, please see comments to review 1 above.

While O₂ (or DMOG) withdrawal may be useful for investigation of cycling O₂ dynamics, unlike HIF protein, the Tomato protein is more stable in normoxia or without DMOG. As such the decrease in reporter activity is likely to be a function of transcript levels and protein turnover for the reporter rather than HIF. The high stability of the reporter is advantageous to screening as it allows assaying of reporter output over extended periods after removal of the ligand/stimulation or in processing cells. As addressed in response to Reviewer #1 we have performed experiments in <1% O₂, demonstrating (Figure 3f) similar levels of induction to that of DMOG. We have also discussed above the dynamics of the reporter in comparison to native target gene inductions. We have not performed reoxygenation experiments as this would only provide information of the stability of the Tomato mRNA and protein and not inform on specific or generic HIF target genes/proteins.

Minor:

The figure organization could benefit from moving some of the data from the suppl. figures to the main figures. Lots of data that are required to understand the manuscript are shown in the suppl. figures while the main figures are rather sparsely in panels.

We agree that space limitations and the scale of the paper makes some of the described text difficult to follow due to substantial supplementary data. We have attempted to reorganise the figures to move some supplementary material to the main figures (especially in Figure 3). We hope that this will better allow the reviewer and the reader the access to the information needed to understand and interpret the main findings of the paper.

This includes;

FACS 2D GFP/Tomato dotplots moved from Supp to Figure 2.

Low O₂ mcdFLASH-HRE induction, mcdFLASH-PGR dose response curves and drug specificity moved to figure 3 from Supp.

Reviewer #3 (Remarks to the Author):

dFLASH; Dual Fluorescent transcription factor Activity Sensor for Histone integrated live-cell reporting and high-content screening

Comments to the authors

Allen et al report the development of dFLASH platform that is a modular transcription factor sensor that can be used in live cell reporting for multiple downstream

applications such as functional genomics screens, drug discovery and synthetic biology. This system consists of two dual color reporters (an internal control nucEGFP as well as dTomato reporter as a readout). The authors demonstrate the use of this reporter system for readout of endogenous HIF and PGR related pathways. Together these authors developed a system that can be used for a variety of applications. Major strengths of this system are that it can be applied to multiple cell types and the modular design also allows researchers to study various endogenous processes. The dual reporter system also allows readouts through flow cytometry or imaging. Furthermore, they successfully used this system to identify novel inhibitors and activators of the HIF pathway in screens. However, it would be useful if the authors could perhaps address more strongly, the ways in which their system is advantageous or different from previous reporter systems. Alternatively, the authors could provide a comparison of previous reporter systems to better communicate the impact or uniqueness of their system. Second, adding more quantifications to some of the imaging and flow cytometry data could also strengthen the results. Overall the data presented in this manuscript is convincing and the applications of the reporter system are strongly demonstrated. Therefore, it could provide an amenable system for reporter-based screening applications to study other endogenous pathways as well.

We thank the reviewer for the comments and have added some additional quantitation to the flow cytometry experiments in each figure including the % of cells Tomato Induced cells (Figure 2, Figure 3, Supp Figure 5, and Supp Figure 7). The mean fold inductions of populations from imaging or FACS experiment is also described in the monoclonal dFLASH cell lines in Figure 3. We hope that this improves the presentation of the results.

Additional comments:

1. Within the introductions and discussions, the authors could discuss previous systems that were built for TF sensing, for example luciferase or reporter systems have been used in the past to enable HIF1a pathway analysis (See a). Discussion of these systems alongside the one that the authors have built would be useful in communicating the impact of the developed reporter system. Additionally, the authors could also determine how their system addresses some of the shortcomings of the previous systems and the types of pathways that can be analyzed by the system in order to make the discussion more comprehensive.

a. An optimized reporter of the transcription factor hypoxia-factor inducible factor 1a reveals complex HIF-1a activation dynamics in single cells
[https://www.jbc.org/article/S0021-9258\(23\)00241-7/fulltext](https://www.jbc.org/article/S0021-9258(23)00241-7/fulltext)

We believe this is addressed in response to Reviewer #1's comments. However, we specifically address the comments regarding the HIF reporters. Reviewer #3 is correct to point out that there are some other approaches taken to report on 'HIF activity' (<https://doi.org/10.1016/j.cmet.2016.09.015> and <https://doi.org/10.1016/j.jbc.2023.104599>). We have attached a table to compare and contrast some of these systems. However, the protein stability reporter mentioned as well as the other referenced HIF reporters do not specifically report of HIF enhancer activity. The HIF1a reporter in the recent JBC article only reports on O2 dependent HIF protein abundance (not transcriptional activity). Additionally, the latter paper contains a complex hypoxic enhancer + constitutive (SV40 promoter) driven O2

destabilised reporter which could decouple O2 dependent enhancer vs post-translational regulation, potentially complicating outputs. In addition, neither are internally controlled at the same genomic loci. Thus, given that the dFLASH system reports specifically on HIF enhancer activity together with the endogenous oxygen-regulation of the HIF transcription factors, and has an internal control, we believe it is thus more reflective of endogenous HIF signalling and better controlled than already available systems.

- See discussion above

2. In Figure 1B, the authors are showing how their built reporter system can be used for imaging-based analysis. This figure can be strengthened if they include the pixel intensities of the GFP and RFP channels from select regions of the microscopy images. More specifically it would be helpful to see that the RFP increases and the GFP remains constant. It would also help account for the background fluorescence of the Tomato reporter in using the imaging application. Finally, they could also consider adding black/white image panels.

Quantitation of the nuclear Tomato/GFP intensities' are reported in various experiments within the paper as such we have not included mean pixel intensities from images in Figure 1. However, to address reviewers #1 and #3 request for individual Tomato and GFP measurements over time we have included individual Tomato and GFP time course data associated with Figure 3 in supplementary Figure 4. We have modified Figure 1 to black and white image panels as requested .

3. The schematic in Figure 1B is a little misleading, can the authors include more details or enhance the figure to guide the reader? For example, add +DMOG etc. or potentially remove the arrow

We have modified Figure 1 in an attempt to more clearly outline the schematic of the dFLASH system. We hope that this is more useful to the reader.

4. Within the flow plots shown in Figure 2, D-F it would help if the authors quantified the percentage of cells that respond to the given treatments. For example, they can show their gating strategies. They should also comment on the fact that these are not clonal lines and are a heterogeneous population, since they are following up on these experiments with single cells clones.

We have now included quantification for polyclonal populations in dFLASH reporters in Figure 2 and an example of the gating strategy in Extended Data Figure 1. In addition, gating strategy is explained in the Methods section.

- Quantification and Gating strategy and % cells in FACS plots.

a.

b.

Extended Data 1. Representative plots for gating strategies

Representative plots of the gating strategy (see Methods) for vehicle (0.1% DMSO, Top row panels) and ligand treated (bottom row) monoclonal (a) HEK293T mcdFLASH-HRE cells and (b) T47D mcdFLASH-PRE cells. Populations are from Figure 3.

5. Could the authors explain why some of their cells do not express dTomato despite antibiotic selection? A two-axis flow plot with gating may be more comprehensive for the main figure (like those shown in Supp Fig2). For example, their flow plots can indicate a range of fluorescence intensities in the tomato channel, how would these numbers change if the authors gated for the highest GFP expressing cells (where the reporter is integrated in euchromatin region). Would that in turn affect the Tomato readout?).

In polyclonal pools the reporter is integrated primarily into gene bodies due to the Lentiviral integrase interaction with LEDGF and recruitment to H3K36me. While these are generally present at actively transcribed genes, the integration site and thus loci dependent effects on enhancer activity may result. In addition, and more likely, stochastic methylation of the enhancer element and not the constitutive promoter may result in differences in the tomato induction. We view this as more likely given that we have observed that polyclonal cell lines that are selected for a short period of time display more homogenous tomato induction (see Supp Figure 1h). Gating based on the highest GFP does not affect the Tomato induction. This can be visualised in the dot plot flow cytometry of GFP and Tomato in the polyclonal pools. As such we have chosen not to include this in the main or supplementary data of the paper.

6. Figure 4C/Supplementary Fig 6: The authors can include a single cell flow plot and gating information in the figure to make it more comprehensive for the reader (including numerical/gating details e.g. % of cells

We have now included the % induced Tomato and gating as part of Figure 2, Figure 3/Supplementary Fig 4 and Figure 4/Supplementary Fig 6 (Now Supplementary Figure 7). The main text has been updated to reflect this.

7. The authors could clarify the reasoning behind using different promoter versions for CRISPRoffv2.1

SFFVp and EF1a are common choices for constitutive expression of transgenes, SFFV is more compact than the full EF1a promoter but can be more prone to silencing in some cell lines (<https://doi.org/10.1016/j.cels.2022.11.005>). We chose to clone and validate using both of these CRISPRoffv2.1 expressing viruses to provide alternate options for researchers, we have made both of these plasmids available on addgene #207180 and #207181.

8. It appears that the clonal cell lines in Figure 4 produce multiple dTomato peaks, does that indicate that the line is slightly leaky for expression of dTomato? How does it compare to no fluorescence - doing an overlay would be helpful? Could potentially be a challenge for use in screens if there is indeed background.

Figure 4 depicts clonal HEK293T-mcdFLASH-HRE cells which have been infected with lentivirus expressing a guide RNA targeting the promoter of VHL and selected with puromycin and subsequently the CRISPRoffv2.1 lentivirus and selection with blasticidin prior to Flow cytometry. The resultant is a polyclonal cell line expressing the guide RNA and the CRISPRoffv2.1 in the background of the clonal reporter line. This explains the heterogeneous induction of Tomato (albeit in ~60% of cells by 10 days post CRISPRoffv2.1). This is in line with the most successful implementations of CRISPRoffv2.1 in Nunez et al 2021 (<https://doi.org/10.1016/j.cell.2021.03.025>) resulting in 60-80% of cells silencing targeted cell surface genes.

As it is unclear where the reviewer is indeed commenting on the background of unstimulated mcdFLASH-HRE in Supplementary Figure 6, we will also address that here. We find that the reporter displays modest background in the absence of DMOG or O₂ in some cell lines, although we have found this is much less pronounced in U2OS and Hela cell lines that we have since generated (data not shown). As previously reported for the HIF pathway (<https://doi.org/10.1002/jcp.21584> and <https://doi.org/10.1038/sj.onc.1204972>) higher cell density can lead to non-hypoxic stabilisation of the hypoxic inducible factors and this may also play a role in background basal expression of Tomato in certain experiments. We do not expect this to effect the ability to screen for HIF activity using the dFLASH systems.

9. In the CRISPRoff experiment, did the authors also include a non-targeting guide? Might be useful to show or include as control

As PGR is not expressed in HEK293T cells and has not been described to effect the HIF pathway, we have used a sgRNA targeting the promoter of the PGR gene as a negative control. These experiments demonstrate that guide RNAs targeting the VHL promoter specifically induce the expression of the HEK293T-mcdFLASH-HRE reporter. We have altered the main text to reflect this.

e.

10. Not essential but the authors can confirm VHL protein knockdown in cells post CRISPRoff induced silencing using a western blot (shows sensitivity of the system).

VHL is a very well characterised Ub-ligase that is reported to specifically target and degrade HIF1a and HIF2a in normoxia. This is mediated by hydroxylation of two proline residues in the Oxygen dependent degradation domains. Given that it is well-accepted that knockout or mutation of VHL results in stabilisation of HIFa protein and transcriptional output we believe that activation of the mcdFLASH reporter upon CRISPRoffv2.1 knockdown of VHL is sufficient evidence. As such we have not attempted to perform western blots for VHL.

11. Discuss how this reporter system can be of better use compared to current systems for screening methods

Emerging applications in high-throughput pooled or arrayed CRISPR or Drug screens require highly sensitive, consistent, selectable reporters. The dFLASH system provides a modular system that unlike other systems allow single cell normalised quantification of transcriptional outputs in live cells. This allows use in monitoring transcriptional dynamics in single cells, pooled or arrayed CRISPR screens easily engineerable to a variety of cells line and transcription factor response pathways. Unlike luciferase systems, dFLASH allows FACS selectable pools which is commonplace in CRISPR screening. We have rigorously validated the applicability to high throughput screening and describe modular use in specifically reporting on functionally distinct transcription factor pathways.

12. For Figure 5, the authors could describe the Activation Screen and Inhibitor with more detail or clarity within the text such that it is easy to follow, or describe the workflow and expected results with more clarity.

We agree with this feedback and as a result, this section has been reworded and the hit classification has been modified for increased clarity on describing how the dFLASH outputs can be utilised in a HTS manner without substantively changing the conclusions put forward in the earlier version of the manuscript. The workflow is now described with more clarity on Page 9-10, Line 578-625). We have also added a paragraph to the results discussion which now describes how the screen cut offs are

described, with more clarity and uniformity between time points and agonist/antagonist screening. In addition, all relevant screen details are included in Supplementary Table 5 for quick reference to workflow, results and directs the reader to the appropriate methodology as required (attached below).

Category	Parameter	Description
Assay	Type of assay	High Content, Live cell screen of a genetic reporter measuring nuclear TF-dependent Tomato expression and constitutive EGFP expression
	Target	HIF-1 α -dependent Tomato expression
	Primary measurement	Ratio of Tomato expression to EGFP expression.
	Key reagents	HEK293T mcdFLASH-HIF reporter cells and 1mM DMOG in DMSO.
	Assay protocol	Assay protocol can be found in detail in the methods in the sections " High Content Imaging (HCI) " and " Bimodal small molecule screen to identify activators or inhibitors of the hypoxic response pathway. "
	Additional comments	Figure 1 provides a schematic for the genetic reporter.
Library	Library size	1595 compounds supplied at 5mM in 1 μ L of DMSO that were dried onto the plates.
	Library composition	Library was a mixture of synthetic and natural product compounds curated by Prof. Ronald Quinn.
	Source	Compounds were sourced from Compounds Australia (www.compoundsaustralia.com)
	Additional comments	
Screen	Format	96-well plates. Plates supplied were 20 96-well Costar CLS3603 black plates with μ clear bottoms.
	Concentration(s) tested	For Activation screening, 50 μ M for each compound was investigated. For Inhibition screening 25 μ M was investigated.
	Plate controls	At each timepoint, compound-free well with equivalent 0.1% DMSO (negative control) and 1mM DMOG (positive control) were included.
	Reagent/ compound dispensing system	Compounds were dispensed into the 96 well format by Compounds Australia. Cells and DMSO or DMOG were added manually.
	Detection instrument and software	Thermofisher ArrayScan XTI was the imaging instrument. HCS Studio 3.0 was the primary analysis software for detection and quantification of nuclear fluorescence.
	Assay validation/QC	Z' > 0.5 for each screen were confirmed, as was ensuring hits met >2SD parameter for Tomato/GFP in Supplementary Figure 8 .
	Correction factors	N/A
	Normalization	Data was Z score normalized.
	Additional comments	
Post-HTS analysis	Hit criteria	Hit criteria is described in method section " Bimodal small molecule screen to identify activators or inhibitors of the hypoxic response pathway ". For activator screens hits had to be >2SD for Tomato/EGFP and >1SD Tomato MFI Z score while EGFP expression did not change more than 2SD relative to mean of the compound treated population. For inhibitor hits criteria was <2SD for Tomato/EGFP and Tomato MFI, while again EGFP did not change more than 2SD relative to the compound treated population.
	Hit rate	36 hour activator screen: 25 compounds (1.4%) 24 hour activator screen: 8 compounds (0.5%) Overall activator screen: 3 compound replicated between screens (0.18%) 36 hour inhibitor screen: 69 compounds (4.2%) 24 hour inhibitor screen: 81 compounds (5.07%) Overall inhibitor screen: 36 compounds replicated between screens (2.25%).
	Additional assay(s)	Replicate dFLASH High content assays were done at 24 hours on a subset of hits for activator and inhibitor compounds to confirm their activity (Supplementary Figure 9, 10).
	Confirmation of hit purity and structure	Compounds were reordered through Compounds Australia (www.compoundsaustralia.com) to confirm identity prior to re-assay and downstream investigations.
	Additional comments	

13. For Figure 5B/C, the authors could label the top hits from their screen in the plot\
14. For Figure 5B, the squares, circles and triangles can be a little difficult to distinguish, the authors could consider using different colors for clarity.

We agree with the reviewer that these figure panels are difficult to interpret with the current layout. We have therefore aligned all of the data by compound such that it is consistent for each screen and labelled a small number of hits, We have also altered the colouring and symbols used to make it easier to distinguish different compound classifications. Additionally, we added some clarity in the methods as to how these hits were classified (page 10, paragraph 1 line 606-616, **Supplementary Table 5W**).

- 1 Bartoszewski, R. *et al.* Primary endothelial cell-specific regulation of hypoxia-inducible factor (HIF)-1 and HIF-2 and their target gene expression profiles during hypoxia. *FASEB J* **33**, 7929-7941 (2019).
<https://doi.org/10.1096/fj.201802650RR>

Other changes

Minor grammatical and text changes throughout the manuscript

Figure 3

Z' and fold change calculation corrections

Fold Change = 15.33393

Z' = 0.6241246

Response the Round 2 Review

Reviewer #1 (Remarks to the Author):

Overall, the authors have not adequately addressed the issues raised during the initial submission, and the manuscript has not been significantly improved. There does not seem to be a much of an advance here.

- Novelty of the HRE reporter. The authors present a table comparing some prior HIF reporters, which is selective and not accurate. Other HRE-fluorescent reporters exist and have been described over the past 20 years (e.g. PMID 11774035, 17270179, 26598532). Some have dual colors, destabilised fluorescent proteins (PEST or adding the ODDD), and have been shown to work in cell lines and in vivo settings. The HRE-HIF-ODD reporter is not constitutively active (minimal SV40 promoter x3HRE). The addition of the internal fluorescent control remains the only novelty of the reporter, but the published NanoFIRE manuscript now referred to in the authors' response (describing a similar approach to dFLASH but with luciferase) already details the use of the 12XHRE dFLASH fluorescent system.

- The authors state that endogenous enhancers contain 5-6 sites for TFs, but still use 12 HREs. This necessity for 12x HRE is at odds with other prior reporters (e.g. HypoxCR).

- The variability in the inhibitory screens remains a concern. The stability of Tomato is not an advantage as stochastic/basal induction of the reporter will occur (as noted), and dFLASH will not detect turning a TF off. Same argument applies for reoxygenation.

- The assays of cell death or drug toxicity have not been adequately addressed.

- The HRE screen has so far not identified any interesting results.

- The utility of the system in other cell types (SFig. 3D) is not that impressive, with a fairly minimal induction.

Reviewer #2 (Remarks to the Author):

The authors have addressed all concerns in the revised manuscript. Congratulations!

Reviewer #3 (Remarks to the Author):

The authors have successfully addressed all the questions and made modifications to the manuscript accordingly.

We submit that our initial reviewer response addressed all of the reviewers' comments with additional experiments and extensive clarification, noting satisfaction from two of the three reviewers. However, reviewer 1 subsequently raised new criticisms that we were not given the opportunity to address, while re-iterated criticisms seem to result from misinterpretations of some of the data and points

made in the manuscript. Here we seek to clarify concerns of reviewer #1 with this rebuttal and new data demonstrating the performance of the dFLASH system in pooled high-throughput CRISPR screens in two separate cell lines.

We presented what reviewer 1 described as a selective comparison of dFLASH compared to existing platforms because like-for-like empirical comparisons were not possible, largely because in built capabilities in dFLASH are not included in the other systems. This we tried to clearly illustrate in the table which we maintain is comprehensive and accurate. Additionally, reviewer #1 provides no evidence of inaccuracies, which we detail below.

We presented a versatile system, demonstrated by analysing four transcriptional pathways with some associated screening applications. Reviewer 1 focused on nuances of the HIF pathway alone. We disagree that dFLASH does not substantially improve upon previous screening systems as outlined below. We also now provide new data in the manuscript in support of the improved screening potential of the dFLASH system. (**Figure R2** or new **Figure 5** of the manuscript, Page 9 paragraph 3, line 400-423, Page 10 paragraph 10, line 432-448, Page 22 paragraph 1, line 862-882)

Novelty of the HRE reporter. The authors present a table comparing some prior HIF reporters, which is selective and not accurate. Other HRE-fluorescent reporters exist and have been described over the past 20 years (e.g. PMID 11774035, 17270179, 26598532). Some have dual colors, destabilised fluorescent proteins (PEST or adding the ODD), and have been shown to work in cell lines and in vivo settings.

We regard our table as complete and accurate and our dFLASH system outperforms the other systems highlighted by reviewer 1 (PMID 11774035, 17270179, 26598532). The improvements in our system include:

- a) incorporating internal controls and selection cassettes
 - b) producing nuclear fluorescent proteins that we demonstrate allows automated nuclear segmentation and quantification of transcriptional responses, a feature absent from PMID 17270179 and 1506559
 - c) being available to other researchers through Addgene and available for benchmarking.
 - d) Unlike the dFLASH reporter system, no other reporter system cited by any reviewer has been shown to function robustly for multiple orthogonal TF response pathways in multiple cell types.
- Importantly, we are unable to find evidence in the literature that there are other internally controlled reporters, conflicting with the statement by reviewer #1. The only HIF reporter with a second readout is detailed in our comparison of reporter systems in the supplementary table and is linked to the cell cycle (PMC4151727). We maintain that this supplementary table comparison of fluorescent reporter systems accurately depicts the advances of our novel, flexible and valuable features of dFLASH.

The HRE-HIF-ODD reporter is not constitutively active (minimal SV40 promoter x3HRE).

Contrary to reviewer 1's claim, HIF reporters containing a 3xHRE-SV40 promoter driven oxygen dependent degraded mCherry^{ODD} (ODD-mCherry) as used in Ortman et al Nature Genetics 2019 CIRSPR screen, has very recently (Posted September 30, 2024.) been shown to produce constitutive activity and poor Hypoxic response element driven transcriptional responses (<https://doi.org/10.1101/2024.09.28.615614> , reproduced below as **Figure R1**) . Note that minimal promoter and response elements adopted in this preprint closely resemble the approach that we have taken with dFLASH. We use an optimised minimal promoter with low background which we have now more clearly outlined in the methods and manuscript (Page 4; line 117, Page 45 line 1198). We hope this more clearly explains the high performance of the dFLASH system in signal induced enhancer activation. This is now evident in the new U2OS-dFLASH-HIF cell line data we have now included which demonstrates a >400 fold induction of Tomato (see below and new Figure 5 of the manuscript, Page 9 paragraph 3, line 400-423, Page 10 paragraph 10, line 432-448, Page 22 paragraph 1, line 862-882,).

[REDACTED]

We have also now additionally benchmarked the performance of the minimal SV40 promoter 3xHRE vs the dFLASH-HRE (2 cell lines) in whole genome CRISPR screening. We demonstrate that the dFLASH system identifies **~>10-15x** more hits (**Figure R2.**, **p<0.05**, and new **Figure 5** of the manuscript, Page 9 paragraph 3, line 400-423, Page 10 paragraph 10, line 432-448, Page 22 paragraph 1, line 862-882, Page 47-49; line 1329-1440) in two separate cell types, including known obligate regulators as top hits (HIF1a and ARNT). In order to benchmark the performance of the dFLASH system in pooled CRISPR screens and compare to orthogonal systems, we have included a new figure into the manuscript demonstrating this (new **Figure 5** of the manuscript, Page 9 paragraph 3, line 400-423, Page 10 paragraph 10, line 432-448, Page 22 paragraph 1, line 862-882, Page 47-49; line 1329-1440). This not only addresses over the request for benchmarking of the dFLASH system and the questioned suitability to high throughput functional genomics screens, but also the ability to sensitively detect turning off transcription factor activity, as outlined by reviewer #1 below.

Figure R2. The number of enriched “hit” genes ($p < 0.01$) from HIF1a whole genome wide CRISPR screens from 3xHRE-sv40- mCherry^{ODD} vs dFLASH-HRE systems. (this is now new **Figure 5** in manuscript, Page 9 paragraph 3, line 400-423, Page 10 paragraph 10, line 432-448, Page 22 paragraph 1, line 862-882, Page 47-49; line 1329-1440)

The addition of the internal fluorescent control remains the only novelty of the reporter, but the published NanoFIRE manuscript now referred to in the authors' response (describing a similar approach to dFLASH but with luciferase) already details the use of the 12XHRE dFLASH fluorescent system.

Our NanoFIRE manuscript (Oct 2023) references the dFLASH system (reference 13 of paper, Biorxiv 2024) and does not detail any of the utilities demonstrated in this paper.

This related publication has no relation to the advances of the dFLASH system or novelty as claimed by reviewer #1, and the methodology and characterisation of the dFLASH system presented here will be of substantial use to researchers attempting to engineer high throughput screening systems.

- The authors state that endogenous enhancers contain 5-6 sites for TFs, but still use 12 HREs. This necessity for 12x HRE is at odds with other prior reporters (e.g. HypoxCR).

High enhancer numbers are not a requirement of the dFLASH system as shown in our other presented reporters (5x GRE and 5xPRE). The literature demonstrates that there is no necessity to use 12 x response elements or any specific number. Our aim was to generate a robust, sensitive and consistent (high Z' score) readout of HIF transcriptional response pathways, not specifically to recapitulate native hypoxic

post-transcriptional cycling. We believe the number of response elements was also sufficiently rebutted in the first review with both synthetic and native examples. The use of multiple, well characterized HIF enhancers (VEGF, LHDA, ENO) as described here is one experimental approach that can be modified according to needs

- The variability in the inhibitory screens remains a concern. The stability of Tomato is not an advantage as stochastic/basal induction of the reporter will occur (as noted), and dFLASH will not detect turning a TF off. Same argument applies for reoxygenation.

We disagree the comments made by the reviewer. The use of an internal control reduces false positives in large screening campaigns and is critical to reduce inevitable false positive hits from interventions that alter basal transcription through cytotoxicity, chromatin effects etc. We have now demonstrated that the dFLASH internal control allowed us to isolate 10 novel inhibitors which were confirmed in biological replicates and dose responses. We addressed in the original rebuttal and in the paper and have now shown further data opposing this comment.

We also note that as described in the rebuttal, dFLASH has been successfully used in a >300,000 HTS where it displayed high Z' values across screening wells ($Z' > 0.75$) (**Figure R3**), where the same strategy of GFP normalization and exclusion of false positives has been employed. We also outline that 2 of the dFLASH systems presented here have been independently validated and used for high throughput screening by Australia's National Drug Discovery Center (NDCC), Walter and Eliza Hall Research Institute (WEHI).

[REDACTED]

Our results indicate that the use of an oxygen destabilized Tomato is not advantageous as reviewer 1 claims. The levels of destabilised tomato reduce rapidly (5-30 mins $T_{1/2}$ as described by numerous groups) upon reoxygenation. This would reduce the signal of a reporter independent of HIF (or ARNT) diminishing the

consistency in screening. This is the opposite of what the reviewer asserts. dFLASH is well suited to inhibition of TF activity and we have demonstrated this in genetic CRISPR screens. We have now included new data in the manuscript in **Figure R2a or Figure 5a of manuscript**, demonstrating that KO of HIF1A in HEK293T or U2OS almost completely blocks reporter induction (also **Figure R2**, new **Figure 5** of the manuscript, Page 9 paragraph 3, line 400-423, Page 10 paragraph 10, line 432-448, Page 22 paragraph 1, line 862-882, Page 47-49; line 1329-1440). We have also demonstrated that KO of the known HIF negative regulator VHL, leads to complete dFLASH activation under normoxia. Stochastic/basal HIF activity is a normal process and very well described in the literature, we see little evidence of this in the dFLASH system unless cells are grown to confluency. We also reiterate from the original rebuttal that the aim of the dFLASH system was not to investigate hypoxic protein dynamics or mimic the native HIF degradation but was to develop a robust and generic TF screening platform which we believe we have demonstrated.

The assays of cell death or drug toxicity have not been adequately addressed.

Destabilised EGFP is present in dFLASH and its loss, as detected by decreased fluorescence, provided a readout of toxicity that was used to identify non-specific effects on transcription and screen out compounds from the chemical library. While this was deemed sufficient for an initial pass during screening, further analyses of nuclear characteristics provided by high content imaging (eg size and shape, possible due to nuclear segmentation provided by dFLASH) are being developed as more subtle indicators of toxicity. We had previously addressed this in our rebuttal and we would like to reiterate that this is not the main focus of the work presented here. In the previously mentioned >300,000 small molecule screen completed at the Australian Drug Discovery Centre, EGFP nuclei count was successfully used to remove non-specific and cytotoxic compounds, highlighting its use as a tool to assess cell death / drug toxicity in a screening setting.

[REDACTED]

- The HRE screen has so far not identified any interesting results.

The work here describes a general reporter platform and not new HIF biology or blockbuster drugs, but instead proof of principle of uses of the dFLASH system. As described and shown above, we have identified a substantial number of new regulators of the HIF pathway from CRISPR screens which will make up a separate manuscript. We now include in the manuscript (**Figure R5c, d**) metadata of screens to outline performance of the dFLASH system in pooled high throughput CRISPR screens, demonstrating the propensity to identify new uncharacterised regulators, and compare them to orthogonal screens.

We also find and describe in this paper 2 new chemical activators, one that works as an iron chelator and several indirect inhibitors. While direct inhibitors of HIF were not detected, this work establishes that the system can efficiently detect regulators of HIF in a high throughput format and that it is suitable for large library screens aimed at discovering direct inhibitors. We reiterate that the aim of this work was to develop a modular, Transcription factor cellular screening platform with a number of high throughput screening applications, which we have demonstrated through application of the dFLASH system to both small molecule and genetic based screens.

- The utility of the system in other cell types (SFig. 3D) is not that impressive, with a fairly minimal induction.

We demonstrate the use of the dFLASH system in a number of cell types, where the majority display robust activation. We have since tested the dFLASH-HRE in HeLa and U2OS cells (**Figure R3**) which further confirm this robust activation. We acknowledge that non-immortalized cells remain a challenge, and a variety of issues may affect the output of the reporters including downstream promoter choice, delivery method etc. These issues are not restricted to the dFLASH system and are shared by many others. We feel that this is a minor aspect of the system which was sufficiently experimentally addressed in the rebuttal and to the satisfaction of 2/3 reviewers. We also reinforce that the dFLASH system was designed with screening applications in mind and not primary cell or in vivo applications, which likely require tailored approaches.

Taken together, we believe that this rebuttal has now addressed all concerns maintained by reviewer #1, including all newly outlined concerns.

Kind Regards,

Dr. David Bersten, on behalf of all authors of the manuscript.

REVIEWERS' COMMENTS

Reviewer #4 (Remarks to the Author):

I was asked primarily to adjudicate the different views of three reviews of this paper and its revision. In my view, the authors have done an excellent job in addressing the comments, and I feel that the comments of Rev 1 are asking too much. Yes, reporters have been developed before including around hypoxia, but the internal control **does** add a lot of value, as does nuclear localization, and the framework appears to be more broadly useful for reporters in general, which to me seem to often be put out into the world in haphazard and poorly controlled ways. I also don't think it's fair to knock the authors for citing their own preprint. The Table that the authors provide now (which I assume will be included in the paper either as main or supplement) is useful and transparent, and to my knowledge accurate (and I say this as the developer of one of the methods listed there, which absolutely has limitations that are fairly noted there). Both imaging and non-imaging based reporter methods need better controls, and the internal control is a terrific way to go. I might suggest that the authors also add scQers (NatureMethods 2024) to the table, which has a sequence-based internal control that I think reinforces the broader point about how important this is in any context where one is looking at single cells rather than bulk (in this case by imaging, in that case by sequencing).

We thank reviewer #4 for the comments and balanced review. We have now included the creative scQrs approach to reporter normalisation into our comparison table and now include it as a Supplementary Data 1.